# Transformers on Markov data: Constant depth suffices

**Nived Rajaraman** *
UC Berkeley

**Marco Bondaschi**
EPFL

**Kannan Ramchandran**
UC Berkeley

**Michael Gastpar**
EPFL

**Ashok Vardhan Makkuva**
EPFL

## Abstract

Attention-based transformers have been remarkably successful at modeling generative processes across various domains and modalities. In this paper, we study the behavior of transformers on data drawn from $k^{\text{th}}$-order Markov processes, where the conditional distribution of the next symbol in a sequence depends on the previous $k$ symbols observed. We observe a surprising phenomenon empirically which contradicts previous findings: when trained for sufficiently long, a transformer with a fixed depth and 1 head per layer is able to achieve low test loss on sequences drawn from $k^{\text{th}}$-order Markov sources, even as $k$ grows. Furthermore, this low test loss is achieved by the transformer's ability to represent and learn the in-context conditional empirical distribution. On the theoretical side, our main result is that a transformer with a single head and three layers can represent the in-context conditional empirical distribution for $k^{\text{th}}$-order Markov sources, concurring with our empirical observations. Along the way, we prove that *attention-only* transformers with $O(\log_2(k))$ layers can represent the in-context conditional empirical distribution by composing induction heads to track the previous $k$ symbols in the sequence. These results provide more insight into our current understanding of the mechanisms by which transformers learn to capture context, by understanding their behavior on Markov sources. Code is available at: https://github.com/Bond1995/Constant-depth-Transformers.

## 1 Introduction

Attention-based transformers have revolutionized the field of natural language processing (NLP) [1, 2] and beyond [3, 4], achieving significant performance gains across tasks like machine translation, text generation, and sentiment analysis. A key factor in their success is their ability to model sequences far more efficiently, and the ability to learn in-context [5, 6].

To understand this capability, a canonical approach is to sample the input from a $k^{th}$-order Markov process, where the next symbol's conditional distribution depends only on the previous $k$ symbols. Recent studies [7, 6, 8] have investigated the ability of transformers to learn Markov processes and establish that learning happens in phases. The transformer eventually learns to represent the conditional $k$-gram model, which is the in-context MLE of the Markov process.

The results in [6, 8] seem to suggest that for low depth transformers to learn Markov processes of order $k$, it is essential that the number of heads scale linearly in $k$. At first glance, this is a bit concerning - real world data generating processes often contain long-range dependencies. How is it that transformers succeed at capturing these kinds of long-range dependencies, while at the same time requiring so many heads to be able to capture the necessary context for $k^{\text{th}}$-order Markov sources?

---

*Correspondence to nived.rajaraman@berkeley.edu.

38th Conference on Neural Information Processing Systems (NeurIPS 2024).

| | Attention-only | | Standard |
|---|---|---|---|
| $L$ | 2 | $\lceil \log_2(k+1) \rceil$ | 3 |
| $H$ | $k$ | 1 | 1 |

$$\cdots \quad \boxed{2} \; \boxed{0} \; \boxed{1} \; \boxed{1} \; \boxed{1} \; \boxed{0} \; \boxed{1} \; \boxed{0} \quad \boxed{X_{n+1}}$$

Figure 1: $k^{\text{th}}$-order Markov process for $k = 4$. The symbol $X_{n+1}$ is sampled from the distribution $P(\cdot|X_n, X_{n-1}, X_{n-2}, X_{n-3})$ which only depends on the last 4 symbols (marked in red).

Table 1: Each column in this table indicates that there is a transformer with $L$ layers and $H$ heads in the first layer which can represent the conditional $k$-gram model.[2]

To understand the nature of this phenomenon, we train low-depth transformers on $k^{\text{th}}$-order Markov sources. These experiments result in two surprising empirical phenomena that seem to contradict previous findings: when trained for sufficiently long, $(i)$ a 2-layer, 1-head transformer can learn $k^{\text{th}}$-order Markov processes for $k$ as large as 4, $(ii)$ a 3-layer, 1-head transformer is able to achieve low test loss on sequences drawn from $k^{\text{th}}$-order Markov sources, even as $k$ grows to be as large as 8 (Fig. 3). In both cases, the values of $k$ for which the models appear to learn $k^{\text{th}}$-order Markov sources are much higher than those predicted in prior experiments [6, 8]. This discrepancy shows that our understanding of the mechanisms used by transformers to learn $k^{\text{th}}$-order Markov processes is not complete and raises a broader question:

> *What is the interplay between depth, number of heads and non-linearity in learning $k^{th}$-order Markov processes?*

In this paper, we approach this question from the point of view of representation power, and provide some partial explanations toward the phenomena illustrated previously.

Our main contributions are as follows:

1. We show, rather surprisingly, that the standard transformer architecture with 3 layers and 1 head per layer is capable of representing the conditional $k$-gram model (Definition 1), and thereby learn $k^{\text{th}}$-order Markov models in-context.

2. Along the way to building up to this result, we consider the simpler family of *attention-only transformers* and show that they can represent the conditional $k$-gram model with $\lceil \log_2(k+1) \rceil$ layers.

3. Under a natural assumption on the nature of the attention patterns learnt by the transformer, we then argue that for $k \geqslant 3$ attention-only transformers *need* at least $\lceil 1 + \log_2(k-2) \rceil$ layers to represent a "$k^{\text{th}}$-order induction head" (Definition 2). Empirically, transformers are observed to learn $k^{\text{th}}$-order induction heads whenever they achieve small test error [6].

The last result is a consequence of a more general tradeoff between the number of layers, $L$, and heads per layer, $H$, an attention-only transformer requires to represent a $k^{\text{th}}$-order induction head, under a natural assumption on the learnt attention patterns. In conjunction, these results also reveal the role of non-linearities (aside from the softmax in the attention) in the transformer architecture. In particular, it appears that layer normalization plays a critical role in the ability of constant-depth transformers to learn the conditional $k$-gram model. Together with the experimental results mentioned previously, these results paint a more comprehensive picture about the representation landscape of transformers in the context of $k^{\text{th}}$-order Markov processes.

**Notation.** Scalars are denoted by italic lower case letters like $x, y$ and Euclidean vectors and matrices in bold $\boldsymbol{x}, \boldsymbol{y}, \boldsymbol{M}$, etc. The notation $\mathbf{0}_{p \times q}$ (resp. $\mathbf{1}_{p \times q}$) refers to the all-zero (resp. all-one) matrix. When it is clear from the context, we omit the dimensions of a matrix. Define $[S] \triangleq \{1, 2, \ldots, S\}$ for $S \in \mathbb{N}$. $\mathbb{I}(\cdot)$ denotes the indicator function and $\text{Unif}(S)$ denotes the uniform distribution over a set $S$.

## 1.1 Related work

There is a large body of active research focused on studying different aspects of transformer models [9, 10, 11, 12]. Our work closely relates to the aspects of understanding the representation power of

---

[2]The requisite embedding dimension and bit-precision to achieve a target additive approximation is discussed in more detail in Sections 4 and 5

transformers, and in-context learning. [13, 14, 15] study the representation capabilities of transformers and show properties such as universal approximation and Turing-completeness. Viewing transformers as sequence to sequence models, [16, 17] study their ability to model formal languages and automata. Along more related lines to our work, [18, 19] present logarithmic depth transformer constructions for representing a $k$-hop generalization of the notion of an induction head [20]. On the other hand the theoretical and mechanistic understanding of in-context learning [21] has received much attention lately [22, 23, 24, 25], focusing on different operating regimes and phases of learning. There are a few recent papers which study the behavior of transformers when trained on data generated from Markov processes, and generalizations thereof [5, 26]. In particular, [7, 8] study the optimization landscape of gradient descent in learning generalizations of Markov processes, and [6] present a study of how transformers learn to represent in-context $k$-gram models, focusing on different phases of learning.

## 2 Preliminaries

We provide the necessary background for Markov processes, the conditional $k$-gram model, and the transformer architecture.

### 2.1 Markov processes

Markov processes are one of the widely used models in sequence modeling [27]. The characterizing property of these processes is that at any time step, the future evolution is only influenced by the most recent states. More formally, a sequence $(X_n)_{n \geqslant 1}$ is a $k^{\text{th}}$-order Markov process on a finite state space $[S]$ with the transition kernel $P$, if surely,

$$P\big(X_{n+1} \mid X_1, \cdots, X_n\big) = P\big(X_{n+1} \mid X_{n-k+1}, \cdots, X_n\big)$$

This property allows us to capture the conditional distribution at any position using only its previous $k$ symbols. This motivates the notion of a conditional $k$-gram, its empirical counterpart, defined for any sequence $(x_1, \ldots, x_n)$.

**Definition 1** (Conditional $k$-gram model). *Given a sequence $(x_1, \cdots, x_n)$ of length $n$ in $[S]^n$, the conditional $k$-gram model $\widehat{\Pr}_k(\cdot \mid x_1, \cdots, x_n)$ corresponds to the in-context estimate of the distribution over symbols conditioned on the last $k$ symbols, i.e. for $x \in [S]$,*

$$\widehat{\Pr}_k(x \mid x_1, \cdots, x_n) \triangleq \frac{\sum_{i=k+1}^n \mathbb{I}(x_i = x, x_{i-1} = x_n, \ldots, x_{i-k} = x_{n-k+1})}{\sum_{i=k+1}^n \mathbb{I}(x_{i-1} = x_n, \ldots, x_{i-k} = x_{n-k+1})},$$

which is defined only so long as the denominator is non-zero. This structure is illustrated in Figure 2a. It is well known that the conditional $k$-gram in Eq. (1) with Laplace smoothing corresponds to the Bayes optimal estimate of the next symbol probability, when the data is drawn from fixed Markov process sampled from a prior distribution [27].

In our experiments, we will consider $k^{\text{th}}$-order Markov kernels sampled from a Dirichlet prior with parameter $\mathbf{1}$. Namely, the transition $P(\cdot|X_1 = i_1, \cdots, X_k = i_k)$ is sampled independently and uniformly on the $S$-dimensional simplex $\Delta_1^S$, for each tuple $(i_1, \cdots, i_k)$.

### 2.2 Transformer architecture

In this paper, we will consider variants of the standard transformer architecture in Figure 2b introduced in [1], with the goal to understand the role of depth and the non-linearities in the architecture. The simplest variant removes all the layer normalization and the (non-linear) feedforward layer, and is referred to as an *attention-only* transformer. The $L$-layer 1-head attention-only transformer with relative position encodings, operating on a sequence of length $T$ is defined in Architecture 1.

The attention scores in layer $\ell$, $\{\text{att}_{n,i}^{(\ell)} : i \leqslant n\}$, are computed as $\texttt{Softmax}\big(\{\big\langle \boldsymbol{W}_K^{(\ell)}(\boldsymbol{x}_j^{(\ell)} + \boldsymbol{p}_{n-j}^{(\ell),K}), \boldsymbol{W}_Q^{(\ell)} \boldsymbol{x}_j^{(\ell)}\big\rangle : j \in [n]\}\big)$. The superscript $(\ell)$ indicates the layer index, and the matrices $\boldsymbol{W}_K^{(\ell)}, \boldsymbol{W}_Q^{(\ell)}, \boldsymbol{W}_V^{(\ell)} \in \mathbb{R}^{d \times d}$ capture the key, query and value matrices in layer $\ell$. Note that the attention-only transformer may include a feedforward layer with linear activations, i.e. a linear

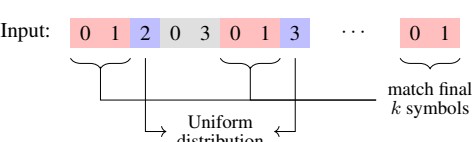

Input: 0 1 2 0 3 0 1 3 ⋯ 0 1

match final $k$ symbols

Uniform distribution

(a) Conditional $k$-gram model. The conditional $k$-gram is the in-context estimate of the Markov process and is realized in two steps. The first step is to find the locations in the sequence (marked red) which match the final $k$ symbols (functionally, a $k^{\text{th}}$-order induction head). The conditional $k$-gram model returns the uniform distribution over the next symbol at these locations (marked blue).

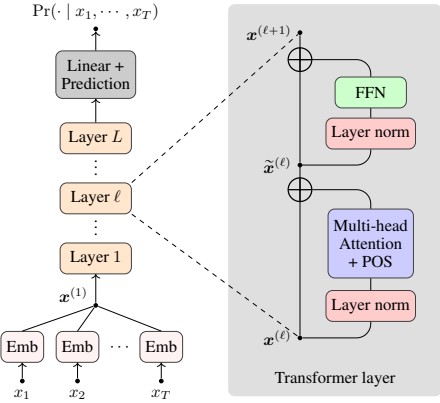

(b) Transformer architecture. POS refers to the relative position encodings.

transformation. For representation purposes, this linear transformation can be combined with the projection matrix in the attention layer, allowing the feedforward layer to be omitted from the model. In the attention layer, we consider relative position encodings (the terms labeled in blue), which translates the key and value vectors depending on the relative position of the embedded symbol.

$$\textbf{for } n = 1, 2, \cdots, T; \; \boldsymbol{x}_n^{(1)} = \texttt{Emb}(x_n) \in \mathbb{R}^d. \qquad \text{(Input embeddings)}$$

$$\textbf{for } \ell = 1, 2, \cdots, L, \textbf{do}$$

$$\quad \textbf{for } n = 1, 2, \cdots, T, \textbf{do}$$

$$\tilde{\boldsymbol{x}}_n^{(\ell)} = \sum_{i \in [n]} \text{att}_{n,i}^{(\ell)} \cdot \boldsymbol{W}_V^{(\ell)} \left( \boldsymbol{x}_i^{(\ell)} + \boldsymbol{p}_{n-i}^{(\ell),V} \right) \in \mathbb{R}^d, \qquad \text{(Attention)}$$

$$\boldsymbol{x}_n^{(\ell+1)} = \boldsymbol{x}_n^{(\ell)} + \tilde{\boldsymbol{x}}_n^{(\ell)}, \qquad \text{(Residual)}$$

$$\rhd \texttt{Here,} \quad \text{att}_{n,i}^{(\ell)} = \texttt{Softmax}_i \left( \left\{ \left\langle \boldsymbol{W}_K^{(\ell)}(\boldsymbol{x}_i^{(\ell)} + \boldsymbol{p}_{n-i}^{(\ell),K}), \boldsymbol{W}_Q^{(\ell)} \boldsymbol{x}_n^{(\ell)} \right\rangle : i \in [S] \right\} \right).$$

$$\text{logit}_T = \boldsymbol{A} \boldsymbol{x}_T^{(L+1)} + \boldsymbol{b} \quad \in \mathbb{R}^S, \qquad \text{(Linear)}$$

$$\text{Pr}_{\boldsymbol{\theta}} \left( \cdot \mid x_1, \cdots, x_T \right) = f \left( \text{logit}_T \right) \in \mathbb{R}^S. \qquad \text{(Prediction)}$$

Architecture 1: Attention-only transformer.

The extension to $H$ heads is straightforward, where in each transformer layer there are $H$ attention layers in parallel, resulting in $\boldsymbol{y}_n^{(\ell,1)}, \cdots, \boldsymbol{y}_n^{(\ell,H)} \in \mathbb{R}^d$ for each $n$. These vectors are concatenated and passed through a linear transformation $\boldsymbol{W}_O^{(\ell)} : \mathbb{R}^{dH} \to \mathbb{R}^d$ which is the output of the attention layer. Finally, the output of the model after $L$ layers is passed through a linear layer, which projects the $d$-dimensional embeddings back into $\mathbb{R}^S$ and the resulting vector is passed through a non-linearity $f$, usually a softmax, to result in the model's prediction of the next symbol probabilities. The theoretical results in this paper will choose $f = \text{ReLU}(\cdot)$.

## 3 Understanding the empirical behavior of transformers

The motivation for the present work comes from a series of experimental results, which challenge our current understanding of transformers in the context of learning Markov processes. Several works in the literature [7, 8, 6] have studied the ability of transformer models to learn $k^{\text{th}}$-order Markov processes. The experimental results present in the literature suggest that in order for a 2 layer transformer model to be able to learn a randomly sampled Markov process of order $k$, it is crucial for the number of heads in the first attention layer to scale linearly with the order, $k$. In particular, the authors of [6] claim that in their experiments, "Single attention headed models could not achieve better performance than bigram (models)" in learning random $k^{\text{th}}$-order Markov processes in-context. Similarly, the

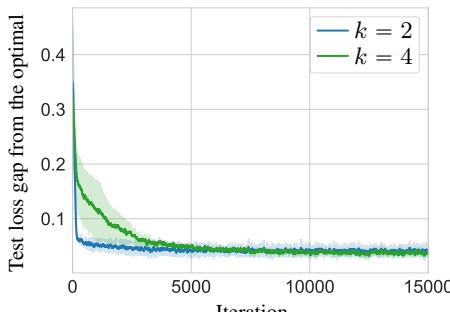 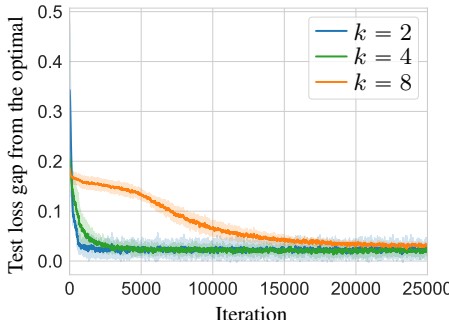

Figure 3: Gap with the optimal test loss for (*a*) a 2-layer, 1-head transformer model (above), and (*b*) a 3-layer, 1-head transformer (below), averaged over 3 runs for each $k$. The models learn the conditional $k$-gram model for randomly sampled $k$-th order Markov processes, even for large $k$.

authors of [8] study a generalization of learning $k^{\text{th}}$-order Markov processes to learning causal processes on degree $k$ graphs. The theory and experiments pertain to 2-layer $k$ head transformers.

In Figure 3, we train 2 and 3-layer transformers with a single head on data drawn from random Markov processes of various orders drawn from a Dirichlet prior. With 2 layers and a single head, we see that the model is able to learn even order-4 Markov processes, and go beyond the simple order-1 processes which were projected to be the limit of its ability to learn. Likewise, with 3 layers, transformers are able to go much further and learn order-8 Markov processes, which was the largest value of $k$ we evaluated on. These results contrast with our current understanding of how induction heads are realized in the parameter space [6, 8] - existing constructions which realize these attention patterns require $k$ heads when the number of layers is 2, and it's unclear how to implement them with fewer heads. At a high level, each of the $k$ heads play a critical role - where, loosely speaking, the $i^{\text{th}}$-head looks back $i$ positions in the sequence.

Building up to our main results, in the sequel, we study the simpler case of attention-only transformers where the feedforward layers and layer normalization are removed.

## 4   Warming up: Attention-only transformers

The study of attention-only transformers trained on Markov processes has garnered some attention in the prior literature. Notably, the authors of [6] study 2-layer 1-head attention-only transformers trained on data drawn from $1^{\text{st}}$-order Markov processes whose parameters are drawn from a Dirichlet prior. The model is observed to learn a very specific behavior, known as an "induction head" [20], which in this setting is able to represent the conditional 1-gram (Eq. (1)).

The induction head mechanism is composed of two layers where the first layer learns the attention pattern $\text{att}_{n,i}^{(1)} = \mathbb{I}(i = n - 1)$, thereby allowing the model to capture information about the symbol at position $n - 1$ in the embedding vector at time $n$. In the second layer, the attention layer picks out those indices $n$ where $x_{n-1} = x_T$, the final symbol in the sequence. At these positions, since $x_{n-1} = x_T$, one would expect that the next symbol $x_n$ is a good predictor of $x_{T+1}$, and the model uses this information to predict the next symbol $x_{T+1}$ according to its conditional empirical estimate, $\widehat{\text{Pr}}_1(x_{T+1}|x_1, \cdots, x_T)$, i.e. the conditional 1-gram model.

**Theorem 1.** *The conditional* 1*-gram model can be represented by a* 2*-layer and* 1*-head attention-only transformer with embedding dimension* $d = 3S + 2$.

Although a version of this result is proved in [6], we include a proof in Appendix A for completeness.

**Remark 1.** *In Theorem 1 and other results to follow, we de-emphasize the role of the bit-precision to which the transformer is implemented. That said, note that when the constructions in Theorems 1 to 3 are implemented to* $O(\log(T))$ *bits of precision, the representation results are realized up to an additive* $O(1/T)$ *error.*

The ideas in Theorem 1 readily extend to representing the conditional $k$-gram model, by instead using $k$ heads in the first layer. The $j^{\text{th}}$ head learns the attention pattern $\text{att}_{n,i}^{(1)} = \mathbb{I}(i = n - j)$ and

concatenating the outputs of the heads, the model learns to aggregate information about $x_n, \cdots, x_{n-k}$ in the embedding vector at time $n$. The second layer realizes what is best described as a "$k^{\text{th}}$-order" induction head, where the model learns to pick out those positions $n$ where for every $j \in [k]$, $x_{n-j} = x_{T-j+1}$, i.e. the history of length $k$ at those positions match the final $k$ symbols in the input sequence ( see Figure 4). This mechanism is also referred to as a long-prefix induction head [28].

**Definition 2** (Higher-order induction head). *A 1-head attention layer is said to realize a $k^{th}$-order induction head if on any sequence $(x_1, \cdots, x_T) \in [S]^T$, for any fixed $n \leqslant T$, as a function of the input sequence, $\mathrm{att}_{n,T}$ is maximized if and only if $x_{n-j} = x_{T-j+1}$ for every $j \in [k]$.*

$k^{\text{th}}$-order induction heads generalize the concept of an induction head [20], and keep track of the positions $i \leqslant n$ where there is a perfect occurrence of the final $k$ symbols in the sequence. Such attention patterns are immediately useful in representing the conditional $k$-gram - increasing the temperature within the softmax of this attention layer results in an attention pattern which converges to the uniform distribution over those positions where the final $k$ symbols $x_{T-k+1}, \cdots, x_T$ are seen previously in the sequence. Loosely, this allows the model to "condition" on the last $k$ symbols in the sequence. With $k$ heads, the model can aggregate information from the previous $k$ positions and implement a $k^{\text{th}}$-order induction head, which leads to the following result. A full proof is discussed in Appendix A.1.

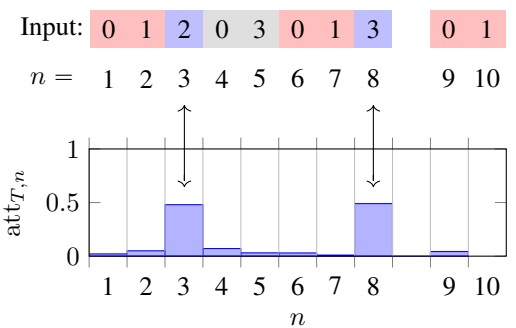

Figure 4: $k^{\text{th}}$-order induction head for $k = 2$. The attention pattern $\mathrm{att}_{T,n}$ is maximized for those values of $n$ at which $x_{T-j+1} = x_{n-j}$ for all $j \in [k]$. These are the positions where the $k$-length prefix at those positions matches with the last $k$ symbols in the sequence.

**Theorem 2.** *The conditional $k$-gram model can be represented by an attention-only transformer with 2 layers, $k$ heads and embedding dimension $d = (k + 2)S + k + 1$.*

While this result is positive, it suggests that a 2-layer transformer requires approximately $k$ times as many parameters to be able to represent the conditional $k$-gram model. The first result we prove is that increasing the depth of the model is exponentially more beneficial, in that a transformer with $O(\log(k))$ depth can estimate in-context $k$-grams.

**Theorem 3.** *The conditional $k$-gram model can be represented by an attention-only transformer with relative position encodings, with $L = \lceil \log_2(k + 1) \rceil$ layers and 1 head per layer. The embedding dimension is $\leqslant 2k(S + 1) + S$.*

With 2 layers and $k$ heads, the transformer aggregates information about each of the previous $k$ positions one step at a time through the $k$ heads. However, with $\Omega(\log(k))$ layers, the same task can be done far more efficiently. In the first attention layer, the model aggregates information about the current and previous position. Namely, using the relative position embeddings, $\boldsymbol{x}_n^{(2)}$ is chosen as a linear combination of $\boldsymbol{x}_n^{(1)} = \mathrm{Emb}(x_n)$ and $\boldsymbol{x}_{n-1}^{(1)} = \mathrm{Emb}(x_{n-1})$. This allows the embedding at position $n$ to aggregate information about $x_n$ and $x_{n-1}$. In the same vein, in the second attention layer, the model aggregates information from $\boldsymbol{x}_n^{(2)}$ and $\boldsymbol{x}_{n-2}^{(2)}$ in $\boldsymbol{x}_n^{(3)}$; the former has information about $x_n$ and $x_{n-1}$, and the latter has information about $x_{n-2}$ and $x_{n-3}$. This expands the "window" of $x_i$'s on which $\boldsymbol{x}_n$ depends on to size 4. In the $\ell^{\text{th}}$ layer, the model aggregates information from $\boldsymbol{x}_n^{(\ell)}$ and $\boldsymbol{x}_{n-2^\ell}^{(\ell)}$ which allows $\boldsymbol{x}_n^{(\ell+1)}$ to effectively depend on the $x_i$'s in a window of size $2^{\ell+1}$ starting at position $n$, namely $x_n, \cdots, x_{n-2^{\ell+1}+1}$. In the final layer, the embedding at position $i$, $\boldsymbol{x}_i^{(L)}$ for $L = \lceil \log_2(k + 1) \rceil$ depends on $x_n, x_{n-1}, \cdots, x_{n-k}$. In the last layer, the model can realize the dot-product $\left\langle \boldsymbol{W}_K^{(L)} \boldsymbol{x}_n^{(L)}, \boldsymbol{W}_Q^{(L)} \boldsymbol{x}_T^{(L)} \right\rangle = \sum_{j=1}^k \mathbb{I}(x_{n-j} = x_{T-j+1})$ by choosing the key and query vectors appropriately. By increasing the temperature in the attention softmax, the attention pattern realized is the uniform distribution on values of $n$ such that $x_{n-j} = x_{T-j+1}$ for every $j \in [k]$, i.e., a $k^{\text{th}}$-order induction head. The full proof of this result is provided in Appendix B.

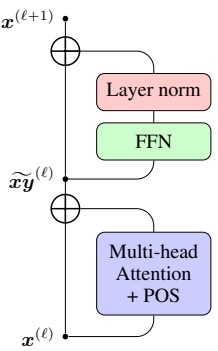

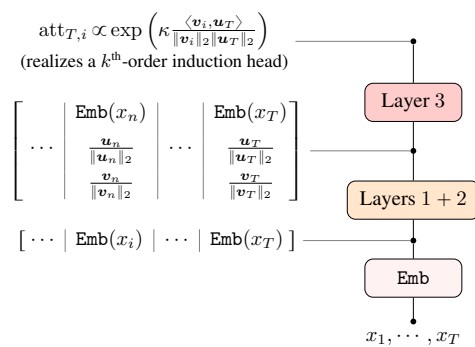

(a) Rearranged transformer layer with layer normalization and FFN.

(b) Realizing a $k^{\text{th}}$-order induction head in a 3-layer transformer following the architecture in Figure 5a.

Figure 5: *Disassembling the constant-depth construction.* The first two layers are critical in the model's ability to capture information from the previous $k$ positions. Layer normalization plays a critical role in in the $3^{\text{rd}}$ layer which realizes a $k^{\text{th}}$-order induction head.

While this is a promising step toward understanding the behavior transformers exhibit in Figure 3, showing that depth plays an important role in their ability to represent conditional $k$-gram models, the picture is still not complete. The experimental results in Section 3 do not preclude the possibility that a transformer might not even require logarithmic depth to be able to learn $k^{\text{th}}$-order Markov processes approximately. In the next section, we will study constant-depth transformers and establish a rather surprising positive result about the representation power of this class of models in capturing conditional $k$-grams.

## 5 Understanding the role of non-linearity: Constant-depth constructions

In the previous section, we saw how the transformer uses the power of depth to learn conditional $k$-grams far more efficiently. In particular, every additional attention layer effectively doubles the window of positions $i = n-1, n-2, \cdots$ which the model has access to information about at the current time $n$. By composing $L = \Omega(\log(k))$ attention layers, the model is able to collect enough information within the output embedding $\boldsymbol{x}_n^{(L+1)}$ to be able to realize a $k^{\text{th}}$-order induction head in the next layer. In this section, we prove that adding non-linearity to the architecture, in the form of layer normalization, can significantly change the mechanism in which the transformer realizes this $k^{\text{th}}$-order head. In particular, there are constant depth architectures which allow a $k^{\text{th}}$-order induction head to be realized, surpassing the logarithmic depth attention-only constructions.

**Modification to the standard transformer architecture.** To simplify the proof of our main result, we will consider a subtle modification to the standard transformer architecture, which is presented in Architecture 2 and Figure 5a. We will remove the first layer norm prior to the multi-head attention and move the second layer norm to after the feed-forward network. It is important to note that Theorem 4 holds even for the architecture presented in Figure 2b, which is the architecture we evaluate empirically. The modification we present in Figure 5a allows the construction to be simpler and makes it much easier to convey the key intuition. The main difference compared to the attention-only design presented in Architecture 1 is the addition of layer normalization and a feedforward layer in the for-loop over $n \in [T]$ for each transformer layer $\ell$. The differences between Architectures 2 and 1 are emphasized in blue.

**Theorem 4.** *Conditional $k$-grams can be represented by a transformer with 3 layers, 1 head per layer, relative position encodings and layer normalization. The embedding dimension is $O(S)$.*

**Remark 2.** *Although the proof stated does not bound the approximation error arising from a finite bound on the bit precision of the transformer, in theory, it should suffice to have $\Omega(\log(T) + k)$ bits per parameter for the statement of Theorem 4 to go through with an $O(1/T)$ additive approximation error. The main point is that none of the weights of the model exceed $\exp(k)$ and with $\log(T)$ additional bits per parameter, the approximation error scales as $O(1/T)$.*

$$\widetilde{\boldsymbol{x}}_n^{(\ell)} = \boldsymbol{x}_n^{(\ell)} + \sum_{i \in [n]} \mathrm{att}_{n,i}^{(\ell)} \cdot \boldsymbol{W}_V^{(\ell)} \left( \boldsymbol{x}_i^{(\ell)} + \boldsymbol{p}_{n-i}^{(\ell),V} \right) \in \mathbb{R}^d, \qquad \text{(Attention + Residual}_1\text{)}$$

$$\boldsymbol{y}_n^{(\ell)} = \boldsymbol{W}_2^{(\ell)} \, \mathrm{ReLU} \left( \boldsymbol{W}_1^{(\ell)} \widetilde{\boldsymbol{x}}_n^{(\ell)} \right) \in \mathbb{R}^d, \qquad \text{(FFN)}$$

$$\boldsymbol{l}_n^{(\ell)} = \frac{\boldsymbol{y}_n^{(\ell)} - \mu \mathbf{1}_{d \times 1}}{\sigma} \in \mathbb{R}^d, \qquad \text{(LN)}$$

$$\boldsymbol{x}_n^{(\ell+1)} = \boldsymbol{l}_n^{(\ell)} + \widetilde{\boldsymbol{x}}_n^{(\ell)} \in \mathbb{R}^d, \qquad \text{(Residual}_2\text{)}$$

Architecture 2: Modified transformer architecture. The computations above are carried out for each $n \in [T]$ in each layer $\ell \in [L]$. In the layer normalization step (LN), the feature mean $\mu$ is defined as, $\mathbb{E}_{i \sim \mathrm{Unif}([d])} \left[ \langle e_i^d \boldsymbol{y}_n^{(\ell)} \rangle \right]$ and the feature variance $\sigma^2 = \mathbb{E}_{i \sim \mathrm{Unif}([d])} \left[ \langle e_i^d, \boldsymbol{y}_n^{(\ell)} \rangle^2 \right] - \mu^2$.

### 5.1 Proof sketch

In the attention-only transformer with 2 layers and $k$ heads, the model is able to keep track of where the final $k$ symbols in the sequence appeared previously (i.e., a $k^{\text{th}}$-order induction head) by, loosely, using each head to keep track of the occurrences of one of the final $k$ symbols. On the other hand, with the benefit of more depth, with $L = \Omega(\log(k))$ layers, the model is able to collect enough information within the output embedding $\boldsymbol{x}_n^{(L+1)}$ to be able to realize the same behavior. However, neither of these constructions scale down to the case when the depth and number of heads of the transformer are both constants independent of $k$. We provide a brief sketch of the construction below.

Recall that a $k^{\text{th}}$-order induction head keeps track of the indices $i$ such that $\forall j \in [k]$, $x_{i-j} = x_{n-j+1}$. Defining $\boldsymbol{z}_i \triangleq \sum_{j=1}^{k} 2^j e_{x_{i-j+1}}$, notice that the condition $\{\forall j \in [k], \ x_{i-j} = x_{n-j+1}\}$ can equivalently be captured by writing $\{\boldsymbol{z}_{i-1} = \boldsymbol{z}_n\}$. This true because of the fact that the binary representation of any integer is unique. Furthermore, these vectors, up to scaling, can be realized by softmax attention (namely, $\mathrm{att}_{n,n-i} \propto 2^i$ for $1 \leqslant i \leqslant k$).

With this step, finding occurrences of the last $k$ symbols in the input sequence boils down to realizing an attention pattern in the second layer, $\mathrm{att}_{n,i}^{(2)}$, which is maximized whenever $\boldsymbol{z}_{i-1} = \boldsymbol{z}_n$. While dot-product attention naively encourages those values of $i$ for which $\boldsymbol{z}_{i-1}$ and $\boldsymbol{z}_n$ are "similar" to each other, a qualitative statement is lacking. In general, it will turn out to that a different measure of similarity is necessary within the softmax to be able to encourage those values of $i$ for which these vectors match. This is where the role of layer-normalization comes in.

Instead of the usual dot-product, suppose the attention mechanism in the second layer was,

$$\mathrm{att}_{n,i}^{(2)} \propto \exp \left( -\kappa \left\| \frac{\boldsymbol{z}_{i-1}}{\|\boldsymbol{z}_{i-1}\|_2} - \frac{\boldsymbol{z}_n}{\|\boldsymbol{z}_n\|_2} \right\|_2^2 \right), \qquad (1)$$

where $\kappa$ is the temperature parameter. Then, as the temperature $\kappa$ grows, the attention pattern essentially focuses on those values of $i$ for which $\boldsymbol{z}_i/\|\boldsymbol{z}_{i-1}\|_2 = \boldsymbol{z}_n/\|\boldsymbol{z}_n\|_2$. With this attention pattern, we are thus very close to the statement we wanted to check, $(\boldsymbol{z}_{i-1} \overset{?}{=} \boldsymbol{z}_n)$. As it turns out, for the special structure in the $\boldsymbol{z}_i$'s considered (dyadic sums of one-hot vectors), we may write down,

$$\boldsymbol{z}_{i-1} = \boldsymbol{z}_n \iff \boldsymbol{z}_{i-1}/\|\boldsymbol{z}_{i-1}\|_2 = \boldsymbol{z}_n/\|\boldsymbol{z}_n\|_2.$$

A quantifiable equivalence is provided in Lemma 1.

**Realizing $L_2$-norm attention (eq. (1)).** Observe the equivalence,

$$\left\langle \frac{\boldsymbol{z}_{i-1}}{\|\boldsymbol{z}_{i-1}\|_2}, \frac{\boldsymbol{z}_n}{\|\boldsymbol{z}_n\|_2} \right\rangle = 1 - \frac{1}{2} \left\| \frac{\boldsymbol{z}_{i-1}}{\|\boldsymbol{z}_{i-1}\|_2} - \frac{\boldsymbol{z}_n}{\|\boldsymbol{z}_n\|_2} \right\|_2^2 \qquad (2)$$

Taking a softmax on both sides, notice that the RHS (up to an additive constant) is the $L_2$-norm based attention, while the LHS is the usual dot-product attention between $\boldsymbol{z}_{i-1}/\|\boldsymbol{z}_{i-1}\|_2$ and $\boldsymbol{z}_n/\|\boldsymbol{z}_n\|_2$. Thus on unit-normalized vectors, $L_2$-norm attention and dot product attention are but the same.

While the first layer of the transformer computes the $\boldsymbol{z}_i$'s by a weighted summation, layer normalization fills in the last missing piece of the puzzle which is to normalize them to unit norm. This is a consequence of defining the embedding vectors appropriately, as we discuss more in Appendix C.1.

From this step, realizing the actual conditional $k$-gram model follows readily. In particular, as the temperature $\kappa$ in the attention grows, the attention pattern zooms in on indices $i \in \mathcal{I}_n \triangleq \{k + 1 \leqslant i \leqslant n : \forall j \in [k], x_{i-j} = x_{n-j+1}\}$ in the last layer. The value vectors at this step are the one-hot encoding of $x_i$; putting everything together, the logits realized by the transformer are,

$$\text{logit}_T(x_{T+1}) = \frac{1}{|\mathcal{I}_n|} \sum_{i \in \mathcal{I}_n} \mathbb{I}(x_i = x_T), \tag{3}$$

which is the conditional $k$-gram model (eq. (1)).

While the transformer construction described above only requires two layers, the actual construction we propose differs slightly and has an additional layer. The first two layers of the transformer respectively compute $z_i$ and $z_{i-1}$ which are added to the embedding vector at time $i$. This is important because we need to test whether $z_{i-1} \overset{?}{=} z_n$ and not whether $z_i \overset{?}{=} z_n$ or $z_{i-1} \overset{?}{=} z_n$.

**Summary.** The construction can be summarized as follows: the first layer computes $z_n = \sum_{j=1}^k 2^{j-1} \cdot e_{x_{n-j}}$ by choosing appropriate value vectors and relative position embeddings to realize the attention pattern $\text{attn}_{n,n-i} \propto 2^i \mathbb{I}(1 \leqslant i \leqslant k)$. The layernorm that follows subsequently can be replaced by RMSnorm, by a simple trick which we discuss in Appendix C.1, resulting in $z_n/\|z_n\|_2$ to be appended to the embedding at time $n$. Using a very similar construction, layer 2 computes $z_{n-1}/\|z_{n-1}\|_2$, which is added to the embedding at time $n$. Finally, in the last layer, the dot-product $\langle z_{i-1}/\|z_{i-1}\|_2, z_n/\|z_n\|_2 \rangle$ defines the attention score, and as the temperature $\kappa$ grows, the pattern converges to $\text{Unif}(\mathcal{I}_n)$. Choosing the value vectors in this layer appropriately gives eq. (3).

# 6  Lower bounds on transformer size

In this section, we study the limits of how shallow a transformer can be made while still capturing conditional $k$-grams. The first result we establish in this vein is a lower bound against 1-layer transformers showing that their expressive power is too limited unless the embedding dimension or number of heads scale near-linearly in $T$.

**Theorem 5.** *Consider any* 1-*layer transformer with layer normalization and feedforward layers, where all the coordinates of the embedding vectors and unnormalized attention scores are computed with $p$ bits of precision. If the transformer is able to compute the conditional* 3-*gram on inputs drawn from $\{0, 1, 2\}^T$ to within an additive error of $1/3T$, then $2pH + dp + 2 \geqslant T/3$.*

Choosing the bit precision to be $p = O(\log(T))$, this implies that for transformers with 1 layer, the sum of the number of heads and the embedding dimension must be at least $\Omega(T/\log(T))$, in order to represent conditional 3-grams to within an additive error of $1/3T$.

## 6.1  Conditional lower bounds on attention-only transformers

While the previous section shows that 1-layer transformers have fairly limited representation power, it is not immediately clear how whether any of these issues are present with transformers with more layers. Indeed, as we discussed in Section 4, an attention-only transformer with $O(\log_2(k))$ layers and 1 head per layer can represent conditional $k$-grams on its input sequences. With the addition of non-linearities, Theorem 4 shows that the model can represent conditional $k$-grams using just a constant number of layers. In this section, we try to understand the gap between these two results and prove conditional lower bounds on the size of attention-only transformers which do not have non-linearities arising from layer normalization.

We prove conditional lower bounds under some natural assumptions on the nature of the attention patterns learnt by the transformer. To motivate these assumptions, consider the experiment in Figure 6, where we train an attention-only transformer with 2 layers and 1 head, on order-1 Markov processes. At test-time, we plot the attention patterns learnt in the first layer of the model on test sequences. Notice that the attention pattern learnt by the model at layer 1 is largely independent of the input sequences themselves and only depends on the position.

**Assumption 1.** *In an L-layer attention-only transformer with $H$ heads per layer, assume that layers $\ell = 1, 2, \cdots, L - 1$ and heads $h \in [H]$ realize an attention pattern where $\text{att}_{n,i}^{(\ell,h)}$ only depends on the positions $n$ and $i$ and on $\ell$ and $h$, but not on the input sequence $x_1, \cdots, x_T$.*

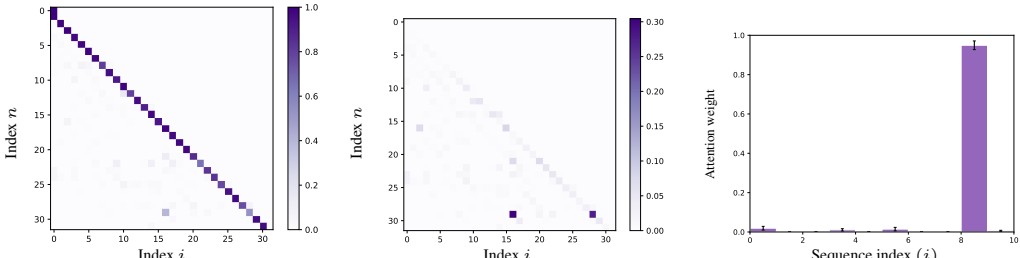

Figure 6: Attention matrix of the first attention layer, for a 2-layer 1-head transformer model trained on an order-1 Markov process, averaged across 100 input sequences of length 128. (a) and (b) plot the mean and standard deviation of the first 32 rows and columns of the attention matrix, while (c) zooms in on the column $n = 10$ and plots the mean attention for this column. (a) and (c) show that for almost all indices $n$, the attention layer focuses only on the previous symbol $x_{n-1}$. (b) shows that the attention pattern does not vary much with the input sequence considered, thereby providing evidence toward Assumption 1. More discussion in Appendix G.

Rather than proving the size lower bound depending on the transformers ability to represent the conditional $k$-gram itself, we consider a simplification and assume that the goal of the model is to represent a $k^{\text{th}}$-order induction head (Definition 2) in the last layer. Although learning a $k^{\text{th}}$-order induction head is not strictly necessary for the transformer to be able to represent conditional $k$-grams, note that every construction we have considered so far (cf. Theorems 1 to 4) go through this mechanism to realize the conditional $k$-gram model. Likewise, for other related problems, such as the causal learning task in [8], the causal structure is captured by an extension of the $k^{\text{th}}$-order induction head to general causal graphs. Our main lower bound is the following result.

**Theorem 6.** *Consider an L-layer transformer with $h_\ell$ heads in layer L. Assuming the transformer satisfies Assumption 1, if $\prod_{\ell=1}^{L-1}(H_\ell + 1) \leqslant k - 2$, the attention pattern in layer L cannot represent a $k^{th}$-order induction head.*

While this lower bound is not unconditional, meaning that it does not directly imply that the transformer cannot represent conditional $k$-grams, it is important to understand the interpretation of this result: attention-only transformers which somehow break through this barrier need to use a significantly different mechanism to realize the conditional $k$-gram model.

Theorem 6 implies that under Assumption 1, a 2-layer attention-only transformer with 1 head cannot realize a $k^{\text{th}}$-order induction head for any $k \geqslant 4$. Likewise, under the same assumption, a 3-layer attention-only transformer with 1 head cannot realize a $k^{\text{th}}$-order induction head for any $k \geqslant 6$. These results give more weight to the experiment in Figure 3 where we observe that a 2-layer transformer learns a $k^{\text{th}}$-order Markov process for $k = 4$ and a 3-layer transformer learns a $k^{\text{th}}$-order Markov process for $k = 8$, and show that non-linearities in the architecture allow the transformer to break past the size barriers in Theorem 4.

## 7 Conclusion

We observe empirically that 2 and 3 layer transformers are able to learn $k^{\text{th}}$-order Markov chains for much higher values of $k$ than previously anticipated. We show there are $O(\log(k))$-layer constructions of attention-only transformers which are able to learn the conditional $k$-gram model, which is the in-context MLE of the Markov model. With non-linearities in the model, we show that a 3-layer 1-head transformer is capable of representing the same. We show that 1-layer transformers cannot represent conditional $k$-grams for any $k \geqslant 3$ unless the number of heads or embedding dimension scale almost linearly in $T$. We also prove a conditional lower bound on the depth and number of heads of attention-only transformers to represent $k^{\text{th}}$-order induction heads, under an assumption on the realized attention patterns.

## Acknowledgments and Disclosure of Funding

The work was partially supported by NSF Grant CCF-2211209 and Swiss NSF Grant 200364.

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

# Appendix

## Table of Contents

**Notation.** The notation $e_i^{d'} \in \mathbb{R}^{d'}$ refers to the one-hot encoding of $i$ in $d'$ dimensions. In other words it is the $i^{\text{th}}$ standard basis vector in $d'$ dimensions. The notation $\texttt{Blkdiag}(\{A_1, A_1, \cdots, A_m\})$ refers to the block diagonal matrix with $i^{\text{th}}$ block as $A_i$.

## A   Proof of Theorem 1

We will first prove Theorem 1. In the first layer, choose the embeddings as,

$$\boldsymbol{x}_n^{(1)} = \texttt{Emb}(x_n) = \kappa \begin{bmatrix} \mathbf{1}_{1\times 2} & e_{x_n}^S & \mathbf{0}_{1\times 2S} \end{bmatrix}^T \in \mathbb{R}^d. \tag{4}$$

for a constant $\kappa > 0$ to be chosen later and $d = 2S + 2$. The relative position encodings will essentially be supported on the first two coordinates, the middle $S$ coordinates are a one-hot encoding of the symbol $x_n$ and the last $2S$ coordinates are $0$. The relative position encodings in the first layer are chosen to be $\boldsymbol{p}_{n-i}^{(1),K} = \kappa\left(-1 + \mathbb{I}(n - i = 1)\right) e_1^d \in \mathbb{R}^d$ and $\boldsymbol{p}_{n-i}^{(1),V} = \mathbf{0} \in \mathbb{R}^d$. Choose $\boldsymbol{W}_K^{(1)}$ and $\boldsymbol{W}_Q^{(1)}$ to be $e_1^d (e_1^d)^T \in \mathbb{R}^{d\times d}$. With this choice,

$$\left\langle \boldsymbol{W}_K^{(1)}\left(\boldsymbol{x}_i^{(1)} + \boldsymbol{p}_{n-i}^{(1),K}\right), \boldsymbol{W}_Q^{(1)}\boldsymbol{x}_n^{(1)} \right\rangle = \kappa\mathbb{I}(n - i = 1) \tag{5}$$

As $\kappa \to \infty$, the attention pattern (which takes the softmax over of these inner products over $i \in [n]$) computes,

$$\text{att}_{n,i}^{(1)} = \mathbb{I}(i = n - 1) \tag{6}$$

for any $n > 1$. Choose the value matrix as,

$$\boldsymbol{W}_V^{(1)} = \begin{bmatrix} \mathbf{0}_{(2+S)\times 2} & \mathbf{0} & \mathbf{0} \\ \mathbf{0} & I_{S\times S} & \mathbf{0} \\ \mathbf{0} & \mathbf{0} & \mathbf{0} \end{bmatrix} \in \mathbb{R}^{d\times d} \tag{7}$$

And with this choice and the residual connection, we get,

$$\boldsymbol{x}_n^{(2)} = \kappa \begin{bmatrix} \mathbf{1}_{1\times 2} & e_{x_n}^S & e_{x_{n-1}}^S & \mathbf{0} \end{bmatrix} \in \mathbb{R}^d \tag{8}$$

which serves as the input to the $2^{\text{nd}}$ transformer layer.

**Layer 2.** In layer 2, the relative position encodings $p_{n-i}^{K,(2)}$ and $p_{n-i}^{V,(2)}$ are all set as 0. The key matrix picks out the $e_{x_n}^S$ block out of $x_n^{(2)}$ and the query vector picks out the $e_{x_{i-1}}^S$ block out of $x_{i-1}^{(2)}$. In particular, these matrices are chosen so that,

$$\begin{aligned}
\boldsymbol{W}_K^{(2)} \boldsymbol{x}_i^{(2)} &= \kappa \begin{bmatrix} \mathbf{1}_{1\times 2} & e_{x_{i-1}}^S & \mathbf{0} \end{bmatrix}^T \in \mathbb{R}^d, \\
\boldsymbol{W}_Q^{(2)} \boldsymbol{x}_n^{(2)} &= \kappa \begin{bmatrix} \mathbf{1}_{1\times 2} & e_{x_n}^S & \mathbf{0} \end{bmatrix}^T \in \mathbb{R}^d
\end{aligned} \tag{9}$$

Taking the inner product of these vectors, and taking $\kappa \to \infty$, observe that the attention pattern concentrates on the uniform distribution over all coordinates $i$ such that $x_{i-1} = x_n$. More formally, the attention pattern for any $n > 1$ is,

$$\text{att}_{n,i}^{(2)} = \frac{\mathbb{I}(x_{i-1} = x_n)}{\sum_{i=2}^n \mathbb{I}(x_{i-1} = x_n)}, \tag{10}$$

assuming $\sum_{i=2}^n \mathbb{I}(x_{i-1} = x_n) > 0$. Having realized this attention pattern, may choose the value and subsequent linear layer appropriately. The value matrix simply picks out the $e_{x_i}^S$ block from $x_i^{(2)}$ and places it into the last $S$ coordinates of $x_i^{(3)}$, and the linear layer simply extracts this block and outputs it (after scaling down by a factor of $\kappa$), realizing the logits,

$$\text{logit}_n = \frac{1}{\sum_{i=2}^n \mathbb{I}(x_{i-1} = x_n)} \sum_{i=2}^n \mathbb{I}(x_{i-1} = x_n) \cdot e_{x_i}^S. \tag{11}$$

if $\sum_{i=2}^n \mathbb{I}(x_{i-1} = x_n) > 0$. In particular, under the same condition,

$$\text{logit}_T(x_{T+1}) = \frac{\sum_{n=2}^T \mathbb{I}(x_n = x_{T+1}, x_{n-1} = x_T)}{\sum_{i=2}^n \mathbb{I}(x_{n-1} = x_T)} \tag{12}$$

assuming $\sum_{i=2}^n \mathbb{I}(x_{n-1} = x_T)$, which is the conditional 1-gram model.

### A.1 Extension to $k$-heads: Proof of Theorem 2

In the first layer, the embeddings are chosen to be,

$$\boldsymbol{x}_n^{(1)} = \text{Emb}(x_n) = \kappa \begin{bmatrix} \mathbf{0}_{1\times k} & | & 1 & | & e_{x_n}^S & | & \mathbf{0}_{1\times(k+1)S} \end{bmatrix}^T \in \mathbb{R}^d \tag{13}$$

With $d = (k+1)(S+1) + S$. The relative position encodings are chosen as $p_i^{K,(1)} = \begin{bmatrix} e_i^k & \mathbf{0} \end{bmatrix}^T$ for $1 \leqslant i \leqslant k$ and $p_i^{K,(1)} = \mathbf{0}$ otherwise. Similarly, $p_i^{V,(1)} = \mathbf{0}$ for every $i$. The $h^{\text{th}}$ head has key and query matrices,

$$\begin{aligned}
\boldsymbol{W}_Q^{(1,h)} &= \begin{bmatrix} \mathbf{0}_{1\times k} & 1 & \mathbf{0} \\ \mathbf{0} & \mathbf{0} & \mathbf{0} \end{bmatrix} \\
\boldsymbol{W}_K^{(1,h)} &= \begin{bmatrix} \mathbf{0}_{1\times(h-1)} & 1 & \mathbf{0} \\ \mathbf{0} & \mathbf{0} & \mathbf{0} \end{bmatrix}
\end{aligned} \tag{14}$$

With these choices, and letting $\kappa \to \infty$, the $h^{\text{th}}$ layer computes the attention pattern,

$$\text{att}_{n,i}^{(1,h)} = \mathbb{I}(i = n - h). \tag{15}$$

Choose the corresponding value matrix as,

$$\boldsymbol{W}_V^{(1,h)} = \begin{bmatrix} \mathbf{0}_{(2+hS)\times 2} & \mathbf{0} & \mathbf{0} \\ \mathbf{0} & I_{S\times S} & \mathbf{0} \end{bmatrix} \tag{16}$$

choosing the projection matrix appropriately, the output of the transformer after the first residual connection is,

$$\boldsymbol{x}_n^{(2)} = \kappa \begin{bmatrix} \mathbf{0}_{1\times k} & | & 1 & | & e_{x_n}^S & | & \cdots & | & e_{x_{n-k}}^S \end{bmatrix}^T. \tag{17}$$

**Layer 2.** In this layer, the relative position encodings $\boldsymbol{p}_{n-i}^{K,(2)}$ and $\boldsymbol{p}_{n-i}^{V,(2)}$ are all set as 0. The key and query matrices are chosen as,

$$
\begin{aligned}
\boldsymbol{W}_Q^{(2)} &= \begin{bmatrix} \mathbf{0}_{Sk \times k} & I_{(Sk+1)\times(Sk+1)} & \mathbf{0} \\ \mathbf{0} & \mathbf{0} & \mathbf{0} \end{bmatrix} \\
\boldsymbol{W}_K^{(2)} &= \begin{bmatrix} \mathbf{0}_{Sk \times (k+S)} & I_{(Sk+1)\times(Sk+1)} \\ \mathbf{0} & \mathbf{0} \end{bmatrix}.
\end{aligned}
\tag{18}
$$

With this choices, we have that,

$$
\left\langle \boldsymbol{W}_K^{(2)} \boldsymbol{x}_i^{(2)}, \boldsymbol{W}_Q^{(2)} \boldsymbol{x}_n^{(2)} \right\rangle = \kappa \sum_{j=1}^{k} \mathbb{I}(x_{i-j} = x_{n-j+1}).
\tag{19}
$$

Taking $\kappa \to \infty$, observe that the attention pattern concentrates on the uniform distribution over all coordinates $i$ such that $x_{i-j} = x_{n-j+1}$ for all $j \in [k]$. More formally, if $\sum_{i=2}^{n} \mathbb{I}(x_{i-1} = x_n) > 0$, the attention pattern for any $n > 1$ is,

$$
\mathrm{att}_{n,i}^{(2)} = \frac{\mathbb{I}(\forall j \in [k],\ x_{i-j} = x_{n-j+1})}{\sum_{i=k+1}^{n} \mathbb{I}(\forall j \in [k],\ x_{i-j} = x_{n-j+1})}.
\tag{20}
$$

The value matrix picks out $e_{x_i}^S$ from the embedding $\boldsymbol{x}_i^{(2)}$ (Equation (17)) and places it in the last $S$ coordinates. The subsequent linear layer picks out the last $S$ coordinates, resulting in the logits,

$$
\mathrm{logit}_n = \sum_{i=k+1}^{n} \frac{\mathbb{I}(\forall j \in [k],\ x_{i-j} = x_{n-j+1})}{\sum_{i=k+1}^{n} \mathbb{I}(\forall j \in [k],\ x_{i-j} = x_{n-j+1})} e_{x_i}^S,
\tag{21}
$$

assuming that $\sum_{i=k+1}^{n} \mathbb{I}(\forall j \in [k],\ x_{i-j} = x_{n-j+1}) > 0$. In particular,

$$
\mathrm{logit}_T(x_{T+1}) = \frac{\sum_{n=k+1}^{T} \mathbb{I}(\forall 0 \leqslant j \leqslant k,\ x_{n-j} = x_{T-j+1})}{\sum_{n=k+1}^{T} \mathbb{I}(\forall 1 \leqslant j \leqslant k,\ x_{n-j} = x_{T-j+1})},
\tag{22}
$$

assuming $\sum_{n=k+1}^{T} \mathbb{I}(\forall 1 \leqslant j \leqslant k,\ x_{n-j} = x_{T-j+1}) > 0$, i.e., the conditional $k$-gram model.

## B    Proof of Theorem 3

Define $k^\star = 2^{\lceil \log_2(k+1) \rceil}$ by rounding $k+1$ up to the nearest power of 2 and $\ell^\star = \log_2(k^\star)$. In the setting of relative position encodings, given the sequence $x_1, \cdots, x_n$, while generating the output of the attention + feedforward layer for the symbol $x_n$, the embeddings $\boldsymbol{x}_i = \mathrm{Emb}(x_n) + \boldsymbol{p}_{n-i}$ are used for $i \in [n]$. In other words, the position encoding vector is taken relative to the end of the sequence, rather than the start of the sequence. Consider the embedding of $x$ as,

$$
\boldsymbol{x}_n^{(1)} = \mathrm{Emb}(x_n) = \begin{bmatrix} \mathbf{0}_{1 \times \ell^\star} & | & 1 & | & e_{x_n}^S & | & \mathbf{0}_{1 \times (k^\star - 1)S} & | & \mathbf{0}_{1 \times S} \end{bmatrix}^T \in \mathbb{R}^{(k^\star+1)S + \ell^\star + 1}
\tag{23}
$$

where $e_i^{d'} \in \mathbb{R}^S$ is the standard basis vector in $d'$ dimensions. And the relative position encoding for the keys as,

$$
\boldsymbol{p}_i^{(1),K} = \begin{cases} \begin{bmatrix} \mathbf{1}_{1 \times \ell^\star} & \mathbf{0} \end{bmatrix}^T, & \text{if } i = 0, \\ \begin{bmatrix} e_{1+\log_2(i)}^{\ell^\star} & \mathbf{0} \end{bmatrix}^T & \text{if } i \in \{1, 2, 4, \cdots, k^\star/2\} \\ \mathbf{0}_{d \times 1} & \text{otherwise.} \end{cases}
\tag{24}
$$

And for the value vectors, $\boldsymbol{p}_i^V = \mathbf{0}$ for all $i$.

For the first layer and first head, we will describe the value, key and query matrices. Choose,

$$
\begin{aligned}
\boldsymbol{W}_K^{(1)} &= \sqrt{\kappa} \begin{bmatrix} 1 & \mathbf{0} \\ \mathbf{0} & \mathbf{0} \end{bmatrix}, \text{ and,} \\
\boldsymbol{W}_Q^{(1)} &= \sqrt{\kappa} \begin{bmatrix} \mathbf{0}_{1 \times l^\star} & 1 & \mathbf{0} \\ \mathbf{0} & \mathbf{0} & \mathbf{0} \end{bmatrix}.
\end{aligned}
\tag{25}
$$

Then, observe that for $i \geqslant 1$,

$$\left\langle W_K^{(1)}\big(x_{n-i} + p_i^{(1),K}\big), W_Q^{(1)} x_n \right\rangle = \kappa \mathbb{I}(i = 1)$$

and for $i = 0$,

$$\left\langle W_K^{(1)}\big(x_n + p_0^{(1),K}\big), W_Q^{(1)} x_n \right\rangle = \kappa$$

In particular, letting $\kappa \to \infty$, the attention pattern is,

$$\text{att}_{n,n-i}^{(1)} = \frac{1}{2}\mathbb{I}(i = 0) + \frac{1}{2}\mathbb{I}(i = 1). \tag{26}$$

Choose the value matrix as,

$$W_V^{(1)} = \begin{bmatrix} \mathbf{0}_{(\ell^\star + S) \times \ell^\star} & \mathbf{0} \\ \mathbf{0} & 2I \end{bmatrix}$$

together with the residual connection, we get,

$$x_n^{(2)} = \begin{bmatrix} \mathbf{0}_{1 \times \ell^\star} & | & 1 & | & e_{x_n}^S & | & e_{x_n}^S + e_{x_{n-1}}^S & | & \mathbf{0}_{1 \times (k^\star - 2)S} & | & \mathbf{0}_{1 \times S} \end{bmatrix}^T \tag{27}$$

**Layer $\ell + 1$.** By induction, assume that the output of the $\ell^{\text{th}}$ transformer layer is of the form,

$$x_n^{(\ell+1)} = \begin{bmatrix} \mathbf{0}_{1 \times \ell^\star} & | & 1 & | & v_n & | & \mathbf{0}_{1 \times (k^\star - 2^\ell)S} & | & \mathbf{0}_{1 \times S} \end{bmatrix}^T \tag{28}$$

for some vector $v_n \in \mathbb{R}^{2^\ell S}$. We will show that with appropriately chosen key, query and value vectors in the $(\ell + 1)^{\text{th}}$ layer, the output of this layer is,

$$x_n^{(\ell+2)} = \begin{bmatrix} \mathbf{0}_{1 \times \ell^\star} & | & 1 & | & v_n & | & v_n + v_{n-2^\ell} & | & \mathbf{0}_{1 \times (k^\star - 2^{\ell+1})S} & | & \mathbf{0}_{1 \times S} \end{bmatrix}^T \tag{29}$$

We will consider the same relative position encodings and query matrix in this layer as in the first layer (Equations (24) and (25)). Consider a key matrix of the form,

$$W_K^{(\ell+1)} = \begin{bmatrix} \mathbf{0}_{1 \times \ell} & \sqrt{\kappa} & \mathbf{0} \\ \mathbf{0} & \mathbf{0} & \mathbf{0} \end{bmatrix}$$

With this choice, observe that for $i \geqslant 1$,

$$\left\langle W_K^{(\ell+1)}\big(x_{n-i}^{(\ell+1)} + p_i^{(\ell+1),K}\big), W_Q^{(\ell+1)} x_n^{(\ell+1)} \right\rangle = \kappa \cdot \mathbb{I}(i = 2^\ell)$$

and for $i = 0$,

$$\left\langle W_K^{(\ell+1)}\big(x_n^{(\ell+1)} + p_0^{(\ell+1),K}\big), W_Q^{(\ell+1)} x_n^{(\ell+1)} \right\rangle = \kappa$$

In particular, letting $\kappa \to \infty$, the attention pattern is,

$$\text{att}_{n,n-i}^{(\ell+1)} = \frac{1}{2}\mathbb{I}(i = 0) + \frac{1}{2}\mathbb{I}(i = 2^\ell). \tag{30}$$

Choosing the value matrix as,

$$W_V^{(\ell+1)} = \begin{bmatrix} \mathbf{0}_{(\ell^\star + 2^\ell S) \times \ell^\star} & \mathbf{0} \\ \mathbf{0} & 2I \end{bmatrix},$$

we get,

$$x_n^{(\ell+2)} = \begin{bmatrix} \mathbf{0}_{1 \times \ell^\star} & | & 1 & | & v_n & | & v_n + v_{n-2^\ell} & | & \mathbf{0}_{1 \times (k^\star - 2^{\ell+1})S} & | & \mathbf{0}_{1 \times S} \end{bmatrix}^T \tag{31}$$

**Final last transformer layer ($\ell = \ell^\star$).** The output of the second last transformer layer, indexed $\ell^\star - 1$ is,

$$z_n^{(\ell^\star)} \triangleq x_n^{(\ell^\star)} = \left[\begin{array}{c|c|c|c|c} \mathbf{0}_{1\times\ell^\star} & 1 & v_n^{(\ell^\star-1)} & v_n^{(\ell^\star-1)} + v_{n-2^{\ell^\star-1}}^{(\ell^\star-1)} & \mathbf{0}_{1\times S} \end{array}\right]^T$$

$$= \left[\begin{array}{c|c|c|c|c} \mathbf{0}_{1\times\ell^\star} & 1 & v_n^{(\ell^\star-1)} & v_n^{(\ell^\star-1)} + v_{n-\frac{k^\star}{2}}^{(\ell^\star-1)} & \mathbf{0}_{1\times S} \end{array}\right]^T,$$

which follows by plugging in the definition of $k^\star$. Note that there exists a linear transformation $L^{(\ell^\star)}$ such that,

$$z_n^{(\ell^\star-1)} \triangleq L^{(\ell^\star)} x_n^{(\ell^\star)} = \left[\begin{array}{c|c|c|c|c} \mathbf{0}_{1\times\ell^\star} & 1 & v_n^{(\ell^\star-1)} & v_{n-\frac{k^\star}{2}}^{(\ell^\star-1)} & \mathbf{0}_{1\times S} \end{array}\right]^T$$

This can be further decomposed as,

$$z_n^{(\ell^\star-1)}$$
$$= \left[\begin{array}{c|c|c|c|c|c} \mathbf{0}_{1\times\ell^\star} & 1 & v_n^{(\ell^\star-2)} & v_n^{(\ell^\star-2)} + v_{n-2^{\ell^\star-2}}^{(\ell^\star-2)} & v_{n-\frac{k^\star}{2}}^{(\ell^\star-2)} & v_{n-\frac{k^\star}{2}}^{(\ell^\star-2)} + v_{n-\frac{k^\star}{2}-2^{\ell^\star-2}}^{(\ell^\star-2)} & \mathbf{0}_{1\times S} \end{array}\right]^T$$

And yet again there exists a linear transformation $L^{(\ell^\star-1)}$ which transforms this as,

$$z_n^{(\ell^\star-2)} \triangleq L^{(\ell^\star-1)} z_n^{(\ell^\star-1)}$$
$$= \left[\begin{array}{c|c|c|c|c|c} \mathbf{0}_{1\times\ell^\star} & 1 & v_n^{(\ell^\star-2)} & v_{n-2^{\ell^\star-2}}^{(\ell^\star-2)} & v_{n-\frac{k^\star}{2}-2^{\ell^\star-2}}^{(\ell^\star-2)} & v_{n-\frac{k^\star}{2}-2^{\ell^\star-2}}^{(\ell^\star-2)} & \mathbf{0}_{1\times S} \end{array}\right]^T$$
$$= \left[\begin{array}{c|c|c|c|c|c} \mathbf{0}_{1\times\ell^\star} & 1 & v_n^{(\ell^\star-2)} & v_{n-\frac{k^\star}{4}}^{(\ell^\star-2)} & v_{n-\frac{k^\star}{2}}^{(\ell^\star-2)} & v_{n-\frac{3k^\star}{4}}^{(\ell^\star-2)} & \mathbf{0}_{1\times S} \end{array}\right]^T \tag{32}$$

By recursing this argument and composing all the linear transformations, up to a global permutation, we get that,

$$\prod_{\ell=1}^{\ell^\star} L^{(\ell)} x_n^{(\ell^\star)} = \left[\begin{array}{c|c|c|c|c|c} \mathbf{0}_{1\times\ell^\star} & 1 & v_n^{(1)} & v_{n-1}^{(1)} & \cdots & v_{n-(k^\star-1)}^{(1)} & \mathbf{0}_{1\times S} \end{array}\right]^T$$

$$= \left[\begin{array}{c|c|c|c|c} \mathbf{0}_{1\times\ell^\star} & 1 & e_{x_n}^S & \cdots & e_{x_{n-(k^\star-1)}}^S & \mathbf{0}_{1\times S} \end{array}\right]^T \tag{33}$$

In the final layer, we will right multiply the key, query and value matrices by $L^\star = \prod_{\ell=1}^{\ell^\star} L^{(\ell)}$. The effect can be interpreted as operating the original key, query and value matrices on the embedding vectors in Equation (33). In the final layer, we will set all the position encodings to be $\mathbf{0}$ and consider the key and query matrices,

$$W_K^{(\ell^\star)} = \sqrt{\kappa} \begin{bmatrix} \mathbf{0}_{Sk\times(\ell^\star+1+S)} & I_{Sk\times Sk} & \mathbf{0} \\ \mathbf{0} & \mathbf{0} & \mathbf{0} \end{bmatrix}$$
$$W_Q^{(\ell^\star)} = \sqrt{\kappa} \begin{bmatrix} \mathbf{0}_{Sk\times(\ell^\star+1)} & I_{Sk\times Sk} & \mathbf{0} \\ \mathbf{0} & \mathbf{0} & \mathbf{0} \end{bmatrix} \tag{34}$$

Then,

$$\left\langle W_K^{(\ell^\star)} L^\star x_{n-i}^{(\ell^\star)}, W_Q^{(\ell^\star)} L^\star x_n^{(\ell^\star)} \right\rangle = \kappa \sum_{j=0}^{k-1} \mathbb{I}(x_{n-j} = x_{i-1-j}) \tag{35}$$

Where we must be careful to note that the input $x_n^{(\ell^\star)}$ contains copies of $e_{x_n}, e_{x_{n-1}}, \cdots, e_{x_{n-k}}$ since $k^\star \geqslant k+1$ by definition.

Letting $\kappa \to \infty$, if there exists $i$ such that $\sum_{j=0}^{k-1} \mathbb{I}(x_{n-j} = x_{i-j-1}) > 0$, for $n \geqslant k$, the attention pattern is,

$$\mathrm{att}_{n,i}^{(\ell^\star)} = \frac{\mathbb{I}(x_{i-1} = x_n, x_{i-2} = x_{n-1}, \cdots, x_{i-k} = x_{n-k+1})}{\sum_{i=k}^{n} \mathbb{I}(x_{i-1} = x_n, x_{i-2} = x_{n-1}, \cdots, x_{i-k} = x_{n-k+1})} \tag{36}$$

Finally, choose,

$$W_V^{(\ell^\star+2)} = \begin{bmatrix} \mathbf{0}_{(d-S)\times(\ell^\star+1)} & \mathbf{0} & \mathbf{0} \\ \mathbf{0}_{S\times(\ell^\star+1)} & I_{S\times S} & \mathbf{0} \end{bmatrix}, \tag{37}$$

we get,

$$\boldsymbol{x}_n^{(\ell^\star+1)} + \sum_{i=k}^n \frac{\mathbb{I}(x_{i-1} = x_n, x_{i-2} = x_{n-1}, \cdots, x_{i-k} = x_{n-k+1})}{\sum_{i=k}^n \mathbb{I}(x_{i-1} = x_n, x_{i-2} = x_{n-1}, \cdots, x_{i-k} = x_{n-k+1})} \begin{bmatrix} \mathbf{0}_{(d-S)\times 1} \\ e_{x_i} \end{bmatrix} \tag{38}$$

Choosing the subsequent linear layer as,

$$\boldsymbol{A} = \begin{bmatrix} \mathbf{0}_{S\times(d-S)} & I_{S\times S} \end{bmatrix} \tag{39}$$

$$\boldsymbol{b} = \mathbf{0}_{S\times 1} \tag{40}$$

Results in the output,

$$\text{logit}_T(x_{T+1}) = \sum_{n=k}^T \frac{\mathbb{I}(x_n = x_{T+1}, x_{n-1} = x_T, x_{n-2} = x_{T-1}, \cdots, x_{n-k} = x_{T-k+1})}{\sum_{n=k}^T \mathbb{I}(x_{n-1} = x_T, x_{n-2} = x_{T-1}, \cdots, x_{n-k} = x_{T-k+1})} \tag{41}$$

which is precisely the in-context conditional $k$-gram.

## C   Proof of Theorem 4

### C.1   Modifying the definition of layer normalization

In every layer, we will perform a simple transformation which is to double the hidden dimension $d$ and add a copy of $-\boldsymbol{x}_n^{(\ell)}$ into the last $d$ coordinates. This is possible by modifying the weights of the transformer appropriately as discussed below. A consequence of this transformation is that the feature mean of the $\boldsymbol{x}_n$'s is $\mu_n = 0$, and therefore the standard deviation $\sigma_n$ simply normalizes by the $L_2$-norm of the features. In order to avoid having to explicitly state this transformation at each layer, we will simply redefine the layer norm LN to output $\boldsymbol{v}/\|\boldsymbol{v}\|_2$ for the input vector $\boldsymbol{v}$, which is realized on the first $d$ coordinates of the transformed embeddings.

This transformation can be realized automatically by redefining the initial embeddings $\text{Emb}(x_n)$, and modifying the weights of the attention and feedforward subnetworks as follows: The input embeddings are changed to $[\text{Emb}(x_n) \quad -\text{Emb}(x_n)]^T \in \mathbb{R}^{2d}$. The key and query matrices are chosen to be 0 on the last $d$ coordinates in every layer; the value matrix for $i \geqslant 1$ is transformed to $\text{Blkdiag}(\{W_V^{(\ell)}, W_V^{(\ell)}\})$, and likewise changing the feedforward layer to the block diagonal matrices $\text{Blkdiag}(\{W_1^{(\ell)}, W_1^{(\ell)}\})$ and $\text{Blkdiag}(\{W_2^{(\ell)}, W_2^{(\ell)}\})$. This transformation adds a copy of $-\boldsymbol{x}_n^{(\ell)}$ into the last $d$ coordinates of the corresponding embeddings.

### C.2   Notation and supplementary lemmas

For each $i \in [T]$, define,

$$\boldsymbol{v}_i = e_{x_{i-1}} + 3 \cdot e_{x_{i-2}} + \cdots + 3^{k-1} \cdot e_{x_{i-k}} \tag{42}$$

$$\boldsymbol{u}_i = e_{x_i} + 3 \cdot e_{x_{i-1}} + \cdots + 3^{k-1} \cdot e_{x_{i-k+1}} \tag{43}$$

Note that although $\boldsymbol{v}_i = \boldsymbol{u}_{i-1}$, we make the distinction between the two to avoid any confusion in what is stored in the embedding vector at time $i$ and at time $i-1$. Furthermore, define,

$$\mathcal{I}_n = \{k+1 \leqslant i \leqslant n : \forall j \in [k], x_{i-j} = x_{n-j+1}\}. \tag{44}$$

**Lemma 1.** *If $i \in \mathcal{I}_n$, $\boldsymbol{z}_i = \boldsymbol{z}_{n-1}$. However, if $i \geqslant k+1$ but $i \notin \mathcal{I}_n$, then, $\left\| \frac{\boldsymbol{v}_i}{\|\boldsymbol{v}_i\|_2} - \frac{\boldsymbol{u}_n}{\|\boldsymbol{u}_n\|_2} \right\|_2 \geqslant 3^{-k}$.*

Let $j^\star \in \{0, 1, \cdots, k-1\}$ denote the largest index $j$ such that $x_{n-j} \neq x_{i-j-1}$. Consider the coordinates $a = x_{n-j^\star} \in [S]$ and $b = x_{i-j^\star-1} \in [S]$. Then,

$$\langle \boldsymbol{v}_n, e_a \rangle - \langle \boldsymbol{u}_i, e_a \rangle \geqslant 3^j - \sum_{j=0}^{j^\star-1} 3^j = \frac{3^{j^\star}}{2}, \tag{45}$$

$$\langle \boldsymbol{u}_i, e_b \rangle - \langle \boldsymbol{v}_n, e_b \rangle \geqslant \frac{3^{j^\star}}{2} \tag{46}$$

If $\|\boldsymbol{v}_n\|_2 \geqslant \|\boldsymbol{u}_i\|_2$, then,

$$\left\langle \frac{\boldsymbol{u}_i}{\|\boldsymbol{u}_i\|_2}, e_b \right\rangle - \left\langle \frac{\boldsymbol{v}_n}{\|\boldsymbol{v}_n\|_2}, e_b \right\rangle \geqslant \frac{\langle \boldsymbol{u}_i, e_b \rangle - \langle \boldsymbol{v}_n, e_b \rangle}{\max\{\|\boldsymbol{u}_i\|_2, \|\boldsymbol{v}_n\|_2\}} \geqslant \frac{3^{j^\star}}{2 \cdot \frac{3^k}{2}} = 3^{j^\star - k} \tag{47}$$

This uses the fact that $\boldsymbol{u}_i$ and $\boldsymbol{v}_n$ are coordinate-wise non-negative. On the other hand, if $\|\boldsymbol{v}_n\|_2 \leqslant \|\boldsymbol{u}_i\|_2$, using a similar analysis,

$$\left\langle \frac{\boldsymbol{u}_i}{\|\boldsymbol{u}_i\|_2}, e_a \right\rangle - \left\langle \frac{\boldsymbol{v}_n}{\|\boldsymbol{v}_n\|_2}, e_a \right\rangle \geqslant 3^{j^\star - k}. \tag{48}$$

In either case, there is a coordinate ($a$ or $b$) such that, $\boldsymbol{u}_i/\|\boldsymbol{u}_i\|_2$ and $\boldsymbol{v}_n/\|\boldsymbol{v}_n\|_2$ differ by at least $3^{j^\star - k}$. This implies the lower bound on the $L_2$ norm of the difference of the vectors.

### C.3   Proof of Theorem 4

Choose the input embeddings as,

$$\boldsymbol{x}_n^{(1)} = \mathtt{Emb}(x_n) = \begin{bmatrix} \boldsymbol{0}_{1 \times 3} & e_x^S & \boldsymbol{0}_{1 \times 5S} \end{bmatrix}^T \in \mathbb{R}^{6S+3} \tag{49}$$

In the first two layers we will use the same relative position embeddings, in particular,

$$\boldsymbol{p}_i^{(1),K} = \boldsymbol{p}_i^{(2),K} = \begin{cases} \sqrt{\log(3)} \cdot \begin{bmatrix} 1 & \boldsymbol{0} \end{bmatrix}^T, & \text{if } i = 0, \\ (i+1)\sqrt{\log(3)} \cdot \begin{bmatrix} 0 & 1 & \boldsymbol{0} \end{bmatrix}^T, & \text{if } i \in \{1, 2, \cdots, k-1\}, , \\ (k+1)\sqrt{\log(3)} \cdot \begin{bmatrix} 0 & 0 & 1 & \boldsymbol{0} \end{bmatrix}^T, & \text{if } i = k. \end{cases} \tag{50}$$

and the value embeddings,

$$\boldsymbol{p}_i^{(1),V} = \boldsymbol{p}_i^{(2),V} = \begin{cases} 3^i \begin{bmatrix} 1 & \boldsymbol{0} \end{bmatrix}^T & \text{for } i \leqslant k \\ \boldsymbol{0} & i > k. \end{cases} \tag{51}$$

In the final layer, we will drop all position-related information and choose $\boldsymbol{p}_i^{(3),K} = \boldsymbol{p}_i^{(3),V} = \boldsymbol{0}$ for all $i$.

**Layer 1.**   Consider the key and query matrices,

$$\boldsymbol{W}_K^{(1)} = \sqrt{\kappa} \cdot \begin{bmatrix} \boldsymbol{1}_{1 \times 2} & \boldsymbol{0} \\ \boldsymbol{0} & \boldsymbol{0} \end{bmatrix}$$
$$\boldsymbol{W}_Q^{(1)} = \sqrt{\kappa} \cdot \begin{bmatrix} \boldsymbol{0}_{1 \times 3} & \boldsymbol{1}_{1 \times S} & \boldsymbol{0} \\ \boldsymbol{0} & \boldsymbol{0} & \boldsymbol{0} \end{bmatrix} \tag{52}$$

Then, observe that,

$$\left\langle \boldsymbol{W}_K^{(1)} \left( \mathtt{Emb}(x_{n-i}) + \boldsymbol{p}_i^{(1),K} \right), \boldsymbol{W}_Q^{(1)} \mathtt{Emb}(x_n) \right\rangle = \kappa(i+1)\log(3) \cdot \mathbb{I}(0 \leqslant i \leqslant \min\{n, k\} - 1)$$

Letting $\kappa \to \infty$, this results in the attention pattern,

$$\mathtt{att}_{n,n-i}^{(1)} = \frac{3^i \mathbb{I}(0 \leqslant i \leqslant \min\{n, k\} - 1)}{\sum_{i'=0}^{\min\{n,k\}-1} 3^{i'}} \tag{53}$$

Choose the value matrix as,

$$\boldsymbol{W}_V^{(1)} = \begin{bmatrix} \boldsymbol{0}_{(S+3) \times 3} & \boldsymbol{0} \\ \boldsymbol{0} & \boldsymbol{I} \end{bmatrix}$$

The output of the attention layer (with the residual connection) is,

$$\widetilde{\boldsymbol{x}}_n^{(1)} = \begin{bmatrix} \boldsymbol{0}_{1 \times 3} & | & e_{x_n}^S & | & \boldsymbol{u}_n & | & \boldsymbol{0}_{1 \times 3S} \end{bmatrix}^T, \text{ where, } \boldsymbol{u}_n = \sum_{i=0}^{\min\{n,k\}-1} \mathtt{att}_{n,n-i} e_{x_{n-i}}^S. \tag{54}$$

In the feedforward layer to follow, we will choose,

$$W_1^{(1)} = I$$

$$W_2^{(1)} = \begin{bmatrix} \mathbf{0}_{(3+2S)\times(3+S)} & \mathbf{0} & \mathbf{0} \\ \mathbf{0} & I_{S\times S} & \mathbf{0} \\ \mathbf{0} & \mathbf{0} & \mathbf{0} \end{bmatrix} \tag{55}$$

Which simply extracts $\boldsymbol{u}_n$ from $\widetilde{\boldsymbol{x}}_n^{(1)}$. With the subsequent layer norm and residual connection, the output of the first layer is,

$$\boldsymbol{x}_n^{(2)} = \begin{bmatrix} \mathbf{0}_{1\times3} & \big| & e_{x_n}^S & \big| & \boldsymbol{u}_n & \big| & \frac{\boldsymbol{u}_n}{\|\boldsymbol{u}_n\|_2} & \big| & \mathbf{0}_{1\times3S} \end{bmatrix}^T \tag{56}$$

**Layer 2.** In this layer, the relative position encodings and query matrix are the same as in layer 1 but the key matrix is chosen as,

$$W_K^{(2)} = \sqrt{\kappa} \begin{bmatrix} 0 & \mathbf{1}_{1\times2} & \mathbf{0} \\ \mathbf{0} & \mathbf{0} & \mathbf{0} \end{bmatrix} \tag{57}$$

With this choice, observe that,

$$\left\langle W_K^{(2)}(\boldsymbol{x}_{n-i}^{(2)} + \boldsymbol{p}_i^{(1),K}), W_Q^{(2)}\boldsymbol{x}_n^{(2)} \right\rangle = \kappa(i+1)\log(3) \cdot \mathbb{I}(1 \leqslant i \leqslant k) \tag{58}$$

As before, since $\kappa \to \infty$, this results in the attention pattern,

$$\text{att}_{n,n-i}^{(2)} = \frac{3^i \mathbb{I}(1 \leqslant i \leqslant \min\{k, n-1\})}{\sum_{i'=1}^{\min\{k,n-1\}} 3^{i'}} \tag{59}$$

which is similar, but subtly different from the attention pattern in the first layer (Equation (53)). The first layer focuses on indices $n-i$ such that $0 \leqslant i \leqslant k-1$, while this layer focuses on $1 \leqslant i \leqslant k$. Choosing the value and projection matrices as,

$$W_V^{(2)} = \begin{bmatrix} I_{3\times3} & \mathbf{0} & \mathbf{0} \\ \mathbf{0}_{3S\times3} & \mathbf{0} & \mathbf{0} \\ \mathbf{0} & I_{S\times S} & \mathbf{0} \\ \mathbf{0} & \mathbf{0} & \mathbf{0} \end{bmatrix} \tag{60}$$

The output of the attention layer (with the first residual connection) is,

$$\widetilde{\boldsymbol{x}}_n^{(2)} = \begin{bmatrix} Z_n & \big| & \mathbf{0}_{1\times2} & \big| & e_{x_n}^S & \big| & \boldsymbol{u}_n & \big| & \frac{\boldsymbol{u}_n}{\|\boldsymbol{u}_n\|_2} & \big| & \boldsymbol{v}_n & \big| & \mathbf{0}_{1\times2S} \end{bmatrix}^T,$$

$$\text{where, } \boldsymbol{v}_n = \sum_{i=1}^{\min\{k,n-1\}} \text{att}_{n,n-i}\, e_{x_{n-i}}^S, \tag{61}$$

$$\text{and, } Z_n = \sum_{i=1}^{\min\{k,n-1\}} \text{att}_{n,n-i}\, 3^i,$$

It is a short calculation to see that $Z_n = 3^{k+1}/5$ if $n \geqslant k+1$ and otherwise, $Z_n \leqslant 3^k/5$. This will be useful later, since the value of $Z_n$ can be used to determine whether $n \geqslant k+1$ or $n \leqslant k$ which will allow the the next layer to avoid calculating the attention at $i \leqslant k$, where the evaluation $x_n = x_{i-1}, \cdots, x_{n-k+1} = x_{i-k}$ is not well defined. In the subsequent FFN layer, we will choose,

$$W_1^{(2)} = I$$

$$W_2^{(2)} = \begin{bmatrix} \mathbf{0}_{(3+4S)\times(3+3S)} & \mathbf{0} & \mathbf{0} \\ \mathbf{0} & I_{S\times S} & \mathbf{0} \\ \mathbf{0} & \mathbf{0} & \mathbf{0}_{S\times 2S} \end{bmatrix} \tag{62}$$

Which extracts $\boldsymbol{v}_n$ from the embedding $\widetilde{\boldsymbol{x}}_n^{(2)}$. With the layer norm and adding the final residual connection, the output of this layer is,

$$\boldsymbol{x}_n^{(3)} = \begin{bmatrix} Z_n & \big| & \mathbf{0}_{2\times1} & \big| & e_{x_n}^S & \big| & \boldsymbol{u}_n & \big| & \frac{\boldsymbol{u}_n}{\|\boldsymbol{u}_n\|_2} & \big| & \boldsymbol{v}_n & \big| & \frac{\boldsymbol{v}_n}{\|\boldsymbol{v}_n\|_2} & \big| & \mathbf{0}_{S\times1} \end{bmatrix}^T \tag{63}$$

**Layer 3.** In this layer, all the relative position encodings are set as $\mathbf{0}$ and instead,

$$\boldsymbol{W}_Q^{(3)} = \sqrt{2\kappa} \begin{bmatrix} 1 & \mathbf{0} & \mathbf{0} & \mathbf{0} \\ \mathbf{0} & \mathbf{0}_{S\times(2+3S)} & I_{S\times S} & \mathbf{0} \\ \mathbf{0} & \mathbf{0} & \mathbf{0} & \mathbf{0} \end{bmatrix}$$

$$\boldsymbol{W}_K^{(3)} = \sqrt{2\kappa} \cdot \begin{bmatrix} 1 & \mathbf{0} & \mathbf{0} & \mathbf{0} \\ \mathbf{0} & \mathbf{0}_{S\times(2+4S)} & I_{S\times S} & \mathbf{0} \\ \mathbf{0} & \mathbf{0} & \mathbf{0} & \mathbf{0} \end{bmatrix} \tag{64}$$

With these choices,

$$\left\langle \boldsymbol{W}_K^{(3)} \boldsymbol{x}_i^{(3)}, \boldsymbol{W}_Q^{(3)} \boldsymbol{x}_n^{(3)} \right\rangle = 2\kappa Z_i Z_n + \frac{2\kappa \langle \boldsymbol{v}_i, \boldsymbol{u}_n \rangle}{\|\boldsymbol{v}_i\|_2 \cdot \|\boldsymbol{u}_n\|_2}$$

$$= 2\kappa Z_i Z_n + 2\kappa - \kappa \left\| \frac{\boldsymbol{v}_i}{\|\boldsymbol{v}_i\|_2} - \frac{\boldsymbol{u}_n}{\|\boldsymbol{u}_n\|_2} \right\|^2 \tag{65}$$

The resulting attention scores are,

$$\mathrm{att}_{n,i}^{(3)} \propto \exp\left( -\kappa \left\| \frac{\boldsymbol{v}_i}{\|\boldsymbol{v}_i\|_2} - \frac{\boldsymbol{u}_n}{\|\boldsymbol{u}_n\|_2} \right\|^2 + 2\kappa Z_i Z_n \right) \tag{66}$$

Recall that $\mathcal{I}_n = \{k+1 \leqslant i \leqslant n : \forall j \in [k], \, x_{n-j+1} = x_{i-j}\}$. Then for any $i \in \mathcal{I}_n$, $\boldsymbol{v}_i = \boldsymbol{u}_n$, and by Lemma 1, for any $i \geqslant k+1$ but not in $\mathcal{I}_n$,

$$\left\| \frac{\boldsymbol{v}_i}{\|\boldsymbol{v}_i\|_2} - \frac{\boldsymbol{u}_n}{\|\boldsymbol{u}_n\|_2} \right\|_2 \geqslant \frac{1}{3^k}.$$

Note that this gap is small but non-zero. Furthermore, recall that $Z_i = 3^{k+1}/5$ if $i \geqslant k$ and otherwise $Z_i \leqslant 3^k/5$. Thus the attention prefers values of $i$ such that $\boldsymbol{v}_i = \boldsymbol{u}_n$ and such that $i \geqslant k+1$. In particular, as $\kappa \to \infty$, the resulting attention pattern is,

$$\mathrm{att}_{n,:}^{(3)} = \mathrm{Unif}(\mathcal{I}_n). \tag{67}$$

Choosing,

$$\boldsymbol{W}_V^{(3)} = \begin{bmatrix} \mathbf{0} & \mathbf{0} & \mathbf{0} \\ \mathbf{0}_{S\times 3} & I_{S\times S} & \mathbf{0} \end{bmatrix}.$$

We get that,

$$\widetilde{\boldsymbol{x}}_n^{(3)} = \boldsymbol{x}_n^{(3)} + \sum_{i=1}^n \mathrm{att}_{n,i}^{(3)} \begin{bmatrix} \mathbf{0} \\ e_{x_i}^S \end{bmatrix} = \boldsymbol{x}_n^{(3)} + \frac{1}{|\mathcal{I}_n|} \sum_{i \in \mathcal{I}_n} \begin{bmatrix} \mathbf{0} \\ e_{x_i}^S \end{bmatrix}.$$

The feedforward layer is chosen to have $\boldsymbol{W}_1^{(3)} = \boldsymbol{W}_2^{(3)} = \mathbf{0}$, and the overall output of the final transformer layer is therefore just $\widetilde{\boldsymbol{x}}_n^{(3)}$. In the output linear layer, choose,

$$\boldsymbol{A} = \begin{bmatrix} \mathbf{0}_{S\times(d-S)} & I_{S\times S} \end{bmatrix}$$

$$\boldsymbol{b} = \mathbf{0} \tag{68}$$

which results in,

$$\mathrm{logit}_n = \frac{1}{|\mathcal{I}_n|} \sum_{i \in \mathcal{I}_n} e_{x_i}^S = \sum_{i=k+1}^n \frac{\mathbb{I}(\forall 1 \leqslant j \leqslant k, \, x_{i-j} = x_{n-j+1})}{\sum_{i'=k+1}^n \mathbb{I}(\forall 1 \leqslant j \leqslant k, \, x_{i'-j} = x_{n-j+1})} \cdot e_{x_i}$$

In particular,

$$\mathrm{logit}_T(x_{T+1}) = \frac{\sum_{n=k+1}^T \mathbb{I}(\forall 0 \leqslant i \leqslant k, \, x_{n-i} = x_{T-i+1})}{\sum_{n=k+1}^T \mathbb{I}(\forall 1 \leqslant i \leqslant k, \, x_{n-i} = x_{T-i+1})} \tag{69}$$

which is the conditional $k$-gram.

# D  Representation lower bounds for $1$-layer transformers: Proof of Theorem 5

We prove this lower bound by a reduction to communication complexity, and specifically to the set disjointness problem.

Suppose Alice and Bob are given strings $\boldsymbol{a}, \boldsymbol{b} \in \{0, 1\}^n$ which are indicator vectors of sets $A$ and $B$. Their goal is to jointly compute $\mathrm{DIS}(\boldsymbol{a}, \boldsymbol{b}) = \mathbb{I}(\exists i : \boldsymbol{a}_i = \boldsymbol{b}_i = 1)$, which indicates whether $A$ and $B$ intersect or not. Alice and Bob may send a single bit message to the other party over a sequence of communication rounds. The following seminal result by [29] asserts a lower bound on amount of communication required between Alice and Bob to carry out this task.

**Theorem 7** ([29]). *Any deterministic protocol for computing* $\mathrm{DIS}(\boldsymbol{a}, \boldsymbol{b})$ *requires at least n rounds of communication.*

We show that a 1-layer transformer with sufficiently small embedding dimension / number of heads can be used to simulate a two-way communication protocol between Alice and Bob to solve $\mathrm{DIS}(\boldsymbol{a}, \boldsymbol{b})$ in a way which contradicts Yao's lower bound in Theorem 7.

With $m = T/3 - 1$, suppose Alice and Bob have length $m$ bit strings $\boldsymbol{a}, \boldsymbol{b} \in \{0, 1\}^m$. The transformer's input will be a sequence of the form,

$$2, \boldsymbol{a}_1, \boldsymbol{b}_1, 2, \boldsymbol{a}_2, \boldsymbol{b}_2, \cdots, 2, \boldsymbol{a}_m, \boldsymbol{b}_m, 2, 1, \tag{70}$$

of length $3m + 2 = T - 1$. The input basically contains a repeating motif, composed of the symbol 2 followed by one of Alice's bits, and then one of Bob's bits. The last 2 symbols are 2 and 1. We will consider the empirical conditional 3-gram probability the transformer associates with the symbol $x_T = 2$. Noting that $x_{T-1} = 1$ and $x_{T-2} = 1$, the conditional 3-gram is computed to be,

$$\frac{\sum_{i=3}^{T-1} \mathbb{I}(x_i = 1, x_{i-1} = 1, x_{i-2} = 2)}{\sum_{i=3}^{T-1} \mathbb{I}(x_{i-1} = 1, x_{i-2} = 2)} \tag{71}$$

Note that if $x_{i-2} = 2$, then $i$ must be of the form $3j$ for $j = 1, \cdots, n$, and we may rewrite the sum as,

$$\frac{\sum_{j=1}^{m} \mathbb{I}(x_{3j} = 1, x_{3j-1} = 1)}{\sum_{j=1}^{m} \mathbb{I}(x_{3j-1} = 1)} = \frac{|A \cap B|}{|B|} \tag{72}$$

Now, let us use the transformer to construct a deterministic communication protocol between Alice and Bob. Alice is given $(x_2, x_5, \cdots, x_{3m-1}) = (\boldsymbol{a}_1, \boldsymbol{a}_2, \cdots, \boldsymbol{a}_m)$ and Bob is given $(x_3, x_6, \cdots, x_{3m}) = (\boldsymbol{b}_1, \boldsymbol{b}_2, \cdots, \boldsymbol{b}_m)$.

In the first round, Alice computes the normalization in the softmax of the attention which comes from the set of inputs she holds. For simplifying notation define,

$$\mathrm{score}^{(h)}(i) = \exp\left(\left\langle \boldsymbol{W}_K^{(h)}(\mathrm{Emb}(x_i) + \boldsymbol{p}_i), \boldsymbol{W}_Q^{(h)}\mathrm{Emb}(x_{T-1}) \right\rangle\right) \tag{73}$$

In particular, for each head $h \in [H]$, she computes,

$$Z_{\mathrm{Alice}}^{(h)} = \log\left(\sum_{j=1}^{m} \mathrm{score}^{(h)}(3j - 1)\right) \tag{74}$$

Assuming that the transformer uses $p$ bits of precision, Alice communicates $Z_{\mathrm{Alice}}^{(h)}$ for each $h$, which corresponds to $pH$ bits of communication. With this information, Bob completes the rest of the normalization term (again up to $p$ bits of precision) and computes,

$$Z^{(h)} = \log\left(Z_{\mathrm{Alice}}^{(h)} + Z_{\mathrm{Bob}}^{(h)} + Z_{\mathrm{common}}^{(h)}\right), \tag{75}$$

$$\text{where } Z_{\mathrm{Bob}}^{(h)} = \log\left(\sum_{j=1}^{m} \mathrm{score}^{(h)}(3j)\right) \tag{76}$$

$$\text{and } Z_{\mathrm{common}}^{(h)} = \log\left(\sum_{j=1}^{m} \mathrm{score}^{(h)}(3j - 2) + \mathrm{score}^{(h)}(T - 2) + \mathrm{score}^{(h)}(T - 1)\right) \tag{77}$$

which is the overall normalization term in the softmax. This is communicated back to Alice, using another $pH$ bits of communication. Next using this information, Alice computes the output of the

attention layer, taking the convex combination corresponding to the inputs she knows. In particular, for each $h \in [H]$ she computes,

$$\sum_{j=1}^{n} \frac{\text{score}^{(h)}(3j-1)}{\exp(Z^{(h)})} \text{Emb}(x_{3j-1}) \in \mathbb{R}^d. \tag{78}$$

across all the heads. Rather than transmitting everything, she concatenates the outputs of the heads, and multiplies them by the value and projection matrices to result in the output $\boldsymbol{y}_{\text{Alice}}$ which is $d$-dimensional. This is sent to Bob using $dp$ bits of communication. Subsequently, Bob computes the terms in the attention corresponding to the inputs he knows as well as the public inputs (all the 2's at positions $3j-2$ as well as the last two symbols). In particular,

$$\sum_{j=1}^{m} \frac{\text{score}^{(h)}(3j)}{\exp(Z^{(h)})} \text{Emb}(x_{3j}) + \sum_{j=1}^{m+1} \frac{\text{score}^{(h)}(3j-2)}{\exp(Z^{(h)})} \text{Emb}(2) + \frac{\text{score}^{(h)}(T-1)}{\exp(Z^{(h)})} \text{Emb}(1) \tag{79}$$

These are yet again concatenated across all the heads and multiplied by the value and projection matrices to result in the output $\boldsymbol{y}_{\text{Bob}}$ which is added to $\boldsymbol{y}_{\text{Alice}}$ to result in $\boldsymbol{y}$. Bob passes $\boldsymbol{y}$ through the residual connection, layer norm, and feedforward layers, and subsequently through the linear layer and softmax of the model to result in the output of the model. By assumption, the output of the model approximately captures the conditional 3-gram, which by Equation (72) equals $|A \cap B|/|B|$. Note that if $|A \cap B|/|B|$ is non-zero, it must be at least $1/T$. This means, if the transformer is able to compute the conditional 3-gram to within an additive error of $1/3T$, then Bob can simply threshold the output of the transformer to decide whether $A \cap B = \varnothing$ or not, thereby solving $\text{DIS}(\boldsymbol{a}, \boldsymbol{b})$.

Since this communication protocol is deterministic, by Yao's lower bound in Theorem 7, the number of bits communicated between Alice and Bob must be at least $m = T/3 - 1$. The total number of bits of communication in the protocol is $2pH + dp + 1$ (the last 1 comes from Bob having to communicate the answer to Alice), completing the proof.

# E   Lower bounds on representing $k^{\text{th}}$-order induction heads: Proof of Theorem 6

In this section we prove the size-lower bound on attention-only transformers representing $k^{\text{th}}$-order induction heads in Theorem 6. To enable this result to be better interpreted, we will break it down into two corollaries.

**Corollary 1.** *Consider an $L$-layer attention-only transformer with $1$ head per layer and relative position encodings, which satisfies Assumption 1. If $L \leqslant 1 + \log_2(k-2)$, the attention pattern in layer $L$ of the transformer cannot represent a $k^{\text{th}}$-order induction head.*

**Corollary 2.** *Consider an $2$-layer attention-only transformer with $H$ heads in the first layer and relative position encodings, and assume that Assumption 1 is satisfied. If $H \leqslant k - 3$, the attention pattern in the $2^{\text{nd}}$ layer cannot represent a $k^{\text{th}}$-order induction head.*

We will first prove the result for the case $L = 2$ and $H = 1$, which falls in the intersection of both of these corollaries. We will show that these models cannot represent $k^{\text{th}}$-order induction heads for $k > 3$, under Assumption 1. We subsequently extend it to the general $L$-layer transformer (i.e., Corollary 1) in Appendix E.2 and to the general case with $H_\ell$ heads in layer $\ell \in [L]$ in Appendix E.3.

## E.1   Lower bounds on $2$-layer $1$-head attention-only transformers

In this section we show that under Assumption 1, a 2-layer 1-head attention-only transformer cannot represent $k^{\text{th}}$-order induction heads for any $k \geqslant 4$. We will prove lower bounds on the transformer when the input is binary, i.e., $S = \{0, 1\}$. With relative position embeddings, observe that the first layer of the transformer model learns representations of the form,

$$\boldsymbol{x}_n^{(2)} = \text{Emb}(x_n) + \sum_{i \leqslant n} \text{att}_{n,i}^{(1)} \boldsymbol{W}_V^{(1)} \text{Emb}(x_i) + \sum_{i \leqslant n} \boldsymbol{W}_V^{(1)} \boldsymbol{p}_{n-i}^{V,(1)} \tag{80}$$

where note that the attention pattern only depends on $n$ and $i$ and not on $x_i$ or $x_n$. These representations are input into the second layer, which realizes the attention pattern $\text{att}_{n,i}^{(2)}$, which is proportional

to,

$$\exp\left(\left\langle \boldsymbol{W}_K^{(2)}\big(\boldsymbol{x}_i^{(2)} + \boldsymbol{p}_{n-i}^{K,(2)}\big), \boldsymbol{W}_Q^{(2)}\boldsymbol{x}_n^{(2)}\right\rangle\right). \tag{81}$$

We need this function to be maximized uniquely when $x_{i-1} = x_n, \cdots, x_{i-k} = x_{n-k+1}$. Denoting $\phi(0) = \boldsymbol{W}_V^{(1)}\texttt{Emb}(0)$ and $\phi(1) = \boldsymbol{W}_V^{(1)}\texttt{Emb}(1)$,

$$\boldsymbol{x}_n^{(2)} = \texttt{Emb}(x_n) + \sum_{i \leqslant n} \mathrm{att}_{n,i}^{(1)} \boldsymbol{W}_V^{(1)}\texttt{Emb}(x_i) + \sum_{i \leqslant n} \boldsymbol{W}_V^{(1)}\boldsymbol{p}_{n-i}^{(1),V} \tag{82}$$

$$= x_n\texttt{Emb}(1) + (1 - x_n)\texttt{Emb}(0) + \sum_{i \leqslant n} \mathrm{att}_{n,i}^{(1)}\big(x_i \cdot \phi(1) + (1 - x_i) \cdot \phi(0)\big) + \sum_{i \leqslant n} \boldsymbol{W}_V^{(1)}\boldsymbol{p}_{n-i}^{(1),V} \tag{83}$$

$$= \left(\frac{\texttt{Emb}(1) + \texttt{Emb}(0)}{2} + x_n' \cdot \frac{\texttt{Emb}(1) - \texttt{Emb}(0)}{2}\right) + \sum_{i \leqslant n} \mathrm{att}_{n,i}^{(1)}\left(\frac{\phi(1) + \phi(0)}{2} + x_i' \cdot \frac{\phi(1) - \phi(0)}{2}\right)$$

$$+ \sum_{i \leqslant n} \boldsymbol{W}_V^{(1)}\boldsymbol{p}_{n-i}^{(1),V} \tag{84}$$

where $x_i' \leftarrow 2x_i - 1$. We can write this down as,

$$\boldsymbol{x}_n^{(2)} = \boldsymbol{m}_n^{(1)} + \boldsymbol{M}_n^{(1)} \begin{bmatrix} x_n' & x_{n-1}' & \cdots & x_1' \end{bmatrix}^T \tag{85}$$

where $\boldsymbol{M}_n^{(1)}$ is a matrix of rank at most 2 and of the form,

$$\boldsymbol{M}_n^{(1)} = \left(\frac{\phi(1) - \phi(0)}{2}\right)\begin{bmatrix} \mathrm{att}_{n,n}^{(1)} & \cdots & \mathrm{att}_{n,1}^{(1)} \end{bmatrix} + \left(\frac{\texttt{Emb}(1) - \texttt{Emb}(0)}{2}\right)\begin{bmatrix} 1 & 0 & \cdots & 0 \end{bmatrix} \tag{86}$$

which is independent of $x_1', \cdots, x_n'$. Likewise $\boldsymbol{m}_n^{(1)}$ collects all the vectors in the sum that don't depend on $x_1', \cdots, x_n'$. Now, observe that in the next layer, we wish to show that an induction head cannot be realized by $\mathrm{att}_{n,i}^{(2)}$ for each $i \leqslant n$. We will show this for any value of $i \leqslant n - k$.

In the second layer, we may write down the key vectors as,

$$\boldsymbol{W}_K^{(2)}\left(\boldsymbol{x}_i^{(2)} + \boldsymbol{p}_{n-i}^{(2),K}\right) = \boldsymbol{W}_K^{(2)}\boldsymbol{m}_i^{(1)} + \boldsymbol{W}_K^{(2)}\boldsymbol{M}_i^{(1)} \begin{bmatrix} x_i' & x_{i-1}' & \cdots & x_1' \end{bmatrix}^T + \boldsymbol{W}_K^{(2)}\boldsymbol{p}_{n-i}^{(2),K}. \tag{87}$$

Again, defining the vector $\overline{\boldsymbol{m}}_i^{(1)}$ and the matrix $\overline{\boldsymbol{M}}_i^{(1)}$ appropriately (having rank at most 2), this equals,

$$\overline{\boldsymbol{m}}_i^{(1)}(\{x_i'\} \cup \{x_{-k-1}', \cdots, x_1'\}) + \overline{\boldsymbol{M}}_i^{(1)}\boldsymbol{y} \tag{88}$$

where $\boldsymbol{y} \triangleq \begin{bmatrix} x_{i-1}' & \cdots & x_{i-k}' \end{bmatrix}^T$ and the vector $\overline{\boldsymbol{m}}_i^{(1)}$ depends on $x_i'$ as well as the inputs $x_{i-k-1}', \cdots, x_1'$, which in this context, are treated as nuisance variables since they do not intersect with $\{x_{i-1}', \cdots, x_{i-k}'\} \cup \{x_n, \cdots, x_{n-k+1}\}$. Henceforth we will avoid explicitly stating the dependency of $\overline{\boldsymbol{m}}_i^{(1)}$ on the $x_j$'s. Similarly, the query vector can be written down as,

$$\boldsymbol{W}_Q^{(2)}\boldsymbol{x}_n^{(2)} = \widehat{\boldsymbol{m}}_n^{(1)} + \widetilde{\boldsymbol{M}}_n^{(1)}\boldsymbol{x} + \widehat{\boldsymbol{M}}_n^{(1)}\boldsymbol{y} \tag{89}$$

where $\widehat{\boldsymbol{m}}_n^{(1)}$, $\widetilde{\boldsymbol{M}}_n^{(1)}$ and $\widehat{\boldsymbol{M}}_n^{(1)}$ are defined appropriately, with $\widehat{\boldsymbol{M}}_n^{(1)}$ and $\widehat{\boldsymbol{M}}_n^{(1)}$ of rank at most 2, and $\boldsymbol{x}$ is defined as $\begin{bmatrix} x_n' & \cdots & x_{n-k+1}' \end{bmatrix}^T$. For an appropriate matrix $\boldsymbol{M}_{n,i}^{\times}$, vectors $\boldsymbol{m}_{n,i}^{\times}$ and $\widetilde{\boldsymbol{m}}_{n,i}^{\times}$ and scalar $m_{n,i}^{\times}$, the dot-product of the key and query vectors can be written as,

$$\left\langle \boldsymbol{W}_K^{(2)}\big(\boldsymbol{x}_i^{(2)} + \boldsymbol{p}_{n-i}^{(2),K}\big), \boldsymbol{W}_Q^{(2)}\boldsymbol{x}_n^{(2)}\right\rangle$$

$$= \boldsymbol{x}^T\boldsymbol{M}_{n,i}^{\times}\boldsymbol{y} + \boldsymbol{y}^T\big(\overline{\boldsymbol{M}}_{n,i}^{\times}\big)\boldsymbol{y} + (\boldsymbol{m}_{n,i}^{\times})^T\boldsymbol{x} + (\overline{\boldsymbol{m}}_{n,i}^{\times})^T\boldsymbol{y} + m_{n,i}^{\times} \triangleq f_{n,i}(\boldsymbol{x}, \boldsymbol{y}), \tag{90}$$

Which is a linear function in $\boldsymbol{x}$ and quadratic in $\boldsymbol{y}$, both of which lie on $\{\pm 1\}^k$. Note that the matrix $\boldsymbol{M}_{n,i}^{\times}$ has rank at most 2 since it is a product of $\overline{\boldsymbol{M}}_i^{(1)}$ and $\widetilde{\boldsymbol{M}}_n^{(1)}$, each with rank at most 2. Next we introduce a lemma showing that if $\boldsymbol{M}_{n,i}^{\times}$ is inherently low rank, the quadratic form in Equation (90) which captures the dot-product between the key and value vectors cannot satisfy the property that for every $\boldsymbol{y}$, the function is uniquely maximized at $\boldsymbol{x} = \boldsymbol{y}$. In particular, this means that for any $i \leqslant n - k$, there is some choice of $x_n, x_{n-1}, \cdots, x_{n-k+1}$ such that there are $x_{i-1}, \cdots, x_{i-k}$ such that for at least one $j \in [k]$, $x_{i-j}$ and $x_{n-j-1}$ are not equal, but the attention score is larger than the case when $x_{i-j}$ were equal to $x_{n-j-1}$ for each $j \in [k]$.

**Lemma 2.** *If $M_{n,i}^{\times}$ has rank $\leqslant k-2$, it is impossible for $f_{n,i}(\boldsymbol{x}, \boldsymbol{y})$ to satisfy the property that for every $\boldsymbol{y} \in \{\pm 1\}^k$, the maximizer is uniquely $\boldsymbol{x} = \boldsymbol{y}$.*

The proof is almost complete: if $k \geqslant 4$, then the rank of $M_{n,i}^{\times}$, which is at most 2, does not exceed $k-2$. This means that when $k \geqslant 4$, any attention pattern realized in the second layer must satisfy the property that there exists a string such that the attention is no longer uniquely maximized when $x_n = x_{i-1}, \cdots, x_{n-k+1} = x_{i-k}$.

*Proof.* For the purpose of brevity, define $\mathcal{H}_k = \{\pm 1\}^k$. First consider the reparameterization,

$$\widetilde{\boldsymbol{x}} = \widetilde{\boldsymbol{M}}_{n,i}^{\times}\boldsymbol{x}, \text{ where } \widetilde{\boldsymbol{M}}_{n,i}^{\times} = \begin{bmatrix} (\boldsymbol{M}_{n,i}^{\times})^T \\ (\boldsymbol{m}_{n,i}^{\times})^T \end{bmatrix}. \tag{91}$$

Then, the dot-product of the key and query matrices can be written as,

$$\begin{bmatrix} \boldsymbol{y}^T & 1 \end{bmatrix} \widetilde{\boldsymbol{x}} + \boldsymbol{y}^T (\overline{\boldsymbol{M}}_{n,i}^{\times})\boldsymbol{y} + (\overline{\boldsymbol{m}}_{n,i}^{\times})^T \boldsymbol{y} + m_{n,i}^{\times} \tag{92}$$

Note that this function is linear in $\widetilde{\boldsymbol{x}}$ and therefore must be maximized on a vertex of the convex hull of the domain, $\widetilde{\boldsymbol{M}}_{n,i}^{\times}\mathcal{H}_k \triangleq \{\widetilde{\boldsymbol{M}}_{n,i}^{\times}\boldsymbol{h} : \boldsymbol{h} \in \mathcal{H}_k\}$. If $\boldsymbol{M}_{n,i}^{\times}$ has rank at most $k-2$, the rank of $\widetilde{\boldsymbol{M}}_{n,i}^{\times}$ is at most $k-1$ and cannot be full rank. We show that this must imply that there is a vertex $\boldsymbol{v} \in \mathcal{H}_k$ such that $\widetilde{\boldsymbol{M}}_{n,i}^{\times}\boldsymbol{v}$ is not a unique vertex of the convex hull of $\widetilde{\boldsymbol{M}}_{n,i}^{\times}\mathcal{H}_k$. This means that $\boldsymbol{v}$ cannot be a unique maximizer for $\widetilde{\boldsymbol{x}}$ when maximizing over all strings in Equation (92), and specifically $\boldsymbol{y} = \boldsymbol{v}$ is a witness to Lemma 2.

Below we discuss how to find such a vector $\boldsymbol{v}$. Note that $\widetilde{\boldsymbol{M}}_{n,i}^{\times}$ is not full rank, which implies that there exists a vector $\boldsymbol{n}$ such that $\widetilde{\boldsymbol{M}}_{n,i}^{\times}\boldsymbol{n} = \boldsymbol{0}$. Without loss of generality, let $\boldsymbol{n}_1$ be the smallest non-zero coordinate of $\boldsymbol{n}$ in absolute value. Then the vector $\boldsymbol{n}_1^{-1}\boldsymbol{n}$ has no non-zero coordinates in the interval $(-1, 1)$. We will show that $\text{sign}(\boldsymbol{n}_1^{-1}\boldsymbol{n})$ is a good choice for $\boldsymbol{v}$.

Consider two cases,

**Case I.** Every non-zero coordinate of $\boldsymbol{n}_1^{-1}\boldsymbol{n}$ is in $\{\pm 1\}$. Consider any $\boldsymbol{x} \in \mathcal{H}_k$ which matches with $\boldsymbol{n}$ on the non-zero coordinates. Consider $\boldsymbol{x}'$ which is the same as $\boldsymbol{x}$, except a negation is taken on the coordinates where $\boldsymbol{n}$ is non-zero. Note that $\widetilde{\boldsymbol{M}}_{n,i}^{\times}\boldsymbol{x} = \widetilde{\boldsymbol{M}}_{n,i}^{\times}\boldsymbol{x}'$, for the same value of $\boldsymbol{x}$. This means that for any $\boldsymbol{y}$. In particular, from Equation (92), both $\boldsymbol{x}$ and $\boldsymbol{x}'$ are maximizers, showing that Lemma 2 is true in this case. We circumvent having to find such a vector $\boldsymbol{v}$ in this case.

**Case II.** $\boldsymbol{n}_1^{-1}\boldsymbol{n}$ has non-zero coordinates which are not all in $\{\pm 1\}$. In particular, at least one coordinate where this vector is strictly less than $-1$ or strictly greater than $+1$. In this case, observe that the sign vector $\widetilde{\boldsymbol{n}} = \text{sign}(\boldsymbol{n}_1^{-1}\boldsymbol{n}) \in \mathcal{H}_k$ lies within, but is not a vertex of the convex hull of the set $\mathcal{H}_k \cup \{\boldsymbol{n}_1^{-1}\boldsymbol{n}\}$. The reason for this is simple to see when we assume that $\boldsymbol{n}_1^{-1}\boldsymbol{n}$ has only one coordinate which is not in $[-1, 1]$, say, the coordinate $j = 2$: here, $\widetilde{\boldsymbol{n}}$ can be written down as a convex combination (with non-zero coefficients) of $\boldsymbol{n}_1^{-1}\boldsymbol{n}$ and $\widetilde{\boldsymbol{n}}^{(2)}$; the latter vector is obtained by flipping coordinate 2 of $\widetilde{\boldsymbol{n}}$. When there is more than one coordinate not in $[-1, 1]$, we can peel away these large coordinates in $\boldsymbol{n}_1^{-1}\boldsymbol{n}$ by taking a convex combination of this vector with the vectors $\widetilde{\boldsymbol{n}}^{(j)}$ for the appropriate values of $j$, to return the sign vector $\widetilde{\boldsymbol{n}}$. Here, $\widetilde{\boldsymbol{n}}^{(j)}$ is the version of $\widetilde{\boldsymbol{n}}$ where the $j^{\text{th}}$-coordinate is flipped. This results in the following claim.

**Claim 1.** *The sign vector $\widetilde{\boldsymbol{n}}$ lies within the convex hull of the points $\mathcal{H}_k \cup \{\boldsymbol{n}_1^{-1}\boldsymbol{n}\}$, but is not a vertex of this set.*

In particular, we may write,

$$\widetilde{\boldsymbol{n}} = \alpha_0 \boldsymbol{n}_1^{-1}\boldsymbol{n} + \sum_{j \in [n]} \alpha_j \widetilde{\boldsymbol{n}}^{(j)}. \tag{93}$$

where $\alpha_0 > 0$ and $\sum_{j=0}^{n} \alpha_i = 1$. By left-multiplying this on both sides by $\widetilde{\boldsymbol{M}}_{n,i}^{\times}$ and noting that $\boldsymbol{n}$ lies in the null-space of this matrix, we get,

$$\widetilde{\boldsymbol{M}}_{n,i}^{\times}\widetilde{\boldsymbol{n}} = \sum_{j \in [n]} \alpha_j \widetilde{\boldsymbol{M}}_{n,i}^{\times}\widetilde{\boldsymbol{n}}^{(j)} \tag{94}$$

where note that $\sum_{j\in[n]}\alpha_j$ is strictly less than 1, since $\alpha_0 > 0$. We may write this vector as,

$$\widetilde{M}_{n,i}^{\times}\widetilde{n} = \alpha_0 \mathbf{0} + \sum_{j\in[n]}\alpha_j\widetilde{M}_{n,i}^{\times}\widetilde{n}^{(j)}$$

$$= \frac{\alpha_0}{2^k}\sum_{\boldsymbol{h}\in\mathcal{H}_k}\widetilde{M}_{n,i}^{\times}\boldsymbol{h} + \sum_{j\in[n]}\alpha_j\widetilde{M}_{n,i}^{\times}\widetilde{n}^{(j)} \tag{95}$$

Since $\alpha_0 > 0$, this equation implies that the image of $\widetilde{n}$ under $\widetilde{M}_{n,i}^{\times}$ itself falls within $\mathrm{conv}(\widetilde{M}_{n,i}^{\times}\mathcal{H}_k)$, but is itself not a vertex of this set. This means that $\widetilde{n}$ can never be a maximizer of $f_{n,i}(\cdot,\boldsymbol{y})$ for any $\boldsymbol{y}$, and in particular when $\boldsymbol{y}=\widetilde{n}$, thereby proving Lemma 2. $\qquad\square$

## E.2   $L$-layer attention-only transformers with 1 head per layer: Proof of Corollary 1

*Proof.* The proof largely tracks the 2-layer case, with the main exception that we keep track of how the maximum possible rank of the matrix $M_{n,i}^{\times}$ grows as a function of the depth of the transformer. In the case the 2-layer transformer, we show that it cannot exceed 2. With the addition of more layers, we show that it cannot exceed $2^{L-1}$.

Recall from the notation in Equation (85) that the output of the first attention layer is,

$$\boldsymbol{x}_n^{(2)} = \boldsymbol{m}_n^{(1)} + \boldsymbol{M}_n^{(1)}\begin{bmatrix} x'_n & x'_{n-1} & \cdots & x'_1 \end{bmatrix}^T \tag{96}$$

where $\boldsymbol{M}_n^{(1)}\in\mathbb{R}^{d\times n}$ has rank at most 2. Let us rewrite this as,

$$\boldsymbol{x}_n^{(2)} = \boldsymbol{m}_n^{(1)} + \overline{\boldsymbol{M}}_n^{(1)}\begin{bmatrix} x'_T & x'_{T-1} & \cdots & x'_1 \end{bmatrix}^T \tag{97}$$

where $\boldsymbol{M}_n^{(1)}\in\mathbb{R}^{d\times T}$ is causally masked to be 0's when it operates on $x_i$ for all indices $i > n$. Note that even with this causal masking, $\overline{\boldsymbol{M}}_n^{(1)}$ has rank at most 2, as discussed in Equation (85).

By induction, assume that the output of the $(\ell-1)^{\text{th}}$ attention layer is of the form,

$$\boldsymbol{x}_n^{(\ell)} = \boldsymbol{m}_n^{(\ell-1)} + \overline{\boldsymbol{M}}_n^{(\ell-1)}\boldsymbol{x}_{1:T} \tag{98}$$

where $\boldsymbol{x}_{1:T} \triangleq \begin{bmatrix} x'_T & x'_{T-1} & \cdots & x'_1 \end{bmatrix}^T$. Passing $\boldsymbol{x}_n^{(\ell)}$ through the $\ell^{\text{th}}$ attention layer, we get,

$$\boldsymbol{x}_n^{(\ell+1)} = \boldsymbol{x}_n^{(\ell)} + \sum_{i\leqslant n}\mathrm{att}_{n,i}^{(\ell)}\boldsymbol{W}_V^{(\ell)}\left(\boldsymbol{x}_i^{(\ell)} + \boldsymbol{p}_{n-i}^{(\ell),V}\right) \tag{99}$$

$$= \boldsymbol{m}_n^{(\ell-1)} + \overline{\boldsymbol{M}}_n^{(\ell-1)}\boldsymbol{x}_{1:T} + \sum_{i\leqslant n}\mathrm{att}_{n,i}^{(\ell)}\boldsymbol{W}_V^{(\ell)}\boldsymbol{m}_i^{(\ell-1)} + \sum_{i\leqslant n}\mathrm{att}_{n,i}^{(\ell)}\boldsymbol{W}_V^{(\ell)}\overline{\boldsymbol{M}}_i^{(\ell-1)}\boldsymbol{x}_{1:T}$$

$$+ \sum_{i\leqslant n}\mathrm{att}_{n,i}^{(\ell)}\boldsymbol{W}_V^{(\ell)}\boldsymbol{p}_{n-i}^{(\ell),V} \tag{100}$$

Define,

$$\boldsymbol{m}_n^{(\ell)} = \boldsymbol{m}_n^{(\ell-1)} + \sum_{i\leqslant n}\mathrm{att}_{n,i}^{(\ell)}\boldsymbol{W}_V^{(\ell)}\boldsymbol{m}_i^{(\ell-1)} + \sum_{i\leqslant n}\mathrm{att}_{n,i}^{(\ell)}\boldsymbol{W}_V^{(\ell)}\boldsymbol{p}_{n-i}^{(\ell),V}, \text{ and,} \tag{101}$$

$$\overline{\boldsymbol{M}}_n^{(\ell)} = \overline{\boldsymbol{M}}_n^{(\ell-1)} + \sum_{i\leqslant n}\mathrm{att}_{n,i}^{(\ell)}\boldsymbol{W}_V^{(\ell)}\overline{\boldsymbol{M}}_i^{(\ell-1)} \tag{102}$$

Then, we can write down,

$$\boldsymbol{x}_n^{(\ell+1)} = \boldsymbol{m}_n^{(\ell)} + \overline{\boldsymbol{M}}_n^{(\ell)}\boldsymbol{x}_{1:T} \tag{103}$$

We also inductively assume that for every $i\leqslant n$,

(i) $\overline{\boldsymbol{M}}_i^{(\ell-1)}$ has rank $R \leqslant 2^{\ell-1}$, and,

(ii) $\overline{\boldsymbol{M}}_i^{(\ell-1)}$ can be factorized in the form $\sum_{r=1}^{R}\boldsymbol{u}_r\cdot\boldsymbol{v}_{i,r}^T$, where only the $\boldsymbol{v}_{i,r}$'s depend on $i$, but the $\boldsymbol{u}_r$'s do not depend on $i$.

Both of these conditions are true when $\ell - 1 = 1$ as evidenced by the structure of $M_i^{(1)}$ in Equation (86) and noting that $\overline{M}_i^{(1)}$ is obtained from $M_i^{(1)}$ by right multiplying by a diagonal mask matrix. Using the recursion in Equation (101), we prove that the induction hypotheses $(i)$ and $(ii)$ are true at layer $\ell$ as well. In particular using the decomposition in $(ii)$, observe that,

$$\overline{M}_n^{(\ell)} = \sum_{r=1}^{R} \boldsymbol{u}_r \cdot \boldsymbol{v}_{n,r}^T + \sum_{i \leqslant n} \text{att}_{n,i}^{(\ell)} \boldsymbol{W}_V^{(\ell)} \sum_{r=1}^{R} \boldsymbol{u}_r \cdot \boldsymbol{v}_{i,r}^T \tag{104}$$

$$= \sum_{r=1}^{R} \boldsymbol{u}_r \cdot \boldsymbol{v}_{n,r}^T + \sum_{r=1}^{R} \boldsymbol{W}_V^{(\ell)} \boldsymbol{u}_r \cdot \left( \sum_{i \leqslant n} \text{att}_{n,i}^{(\ell)} \boldsymbol{v}_{i,r} \right)^T \tag{105}$$

$$= \sum_{r=1}^{2R} \boldsymbol{u}_r \cdot \boldsymbol{v}_{n,r}^T \tag{106}$$

where for $r' \in [R]$, $\boldsymbol{u}_{R+r'} \triangleq \boldsymbol{W}_V^{(\ell)} \boldsymbol{u}_r$ and $\boldsymbol{v}_{n,r'} \triangleq \sum_{i \leqslant n} \text{att}_{n,i}^{(\ell)} \boldsymbol{v}_{i,r}$. Since $M_n^{(\ell)}$ is the sum of $2R$ rank 1 matrices and therefore has rank at most $2R \leqslant 2^\ell$, proving both parts of the induction hypothesis.

By induction, at the end of the $(L-1)^{\text{th}}$ layer, we have an output which looks like,

$$\boldsymbol{x}_n^{(L)} = \boldsymbol{m}_n^{(L-1)} + \overline{\boldsymbol{M}}_n^{(L-1)} \boldsymbol{x}_{1:T} \tag{107}$$

where $\boldsymbol{M}_n^{(L-1)}$ has rank at most $2^{L-1}$. More importantly, note that by the causal masking, even though it appears to depend on the whole input sequence through $\boldsymbol{x}_{1:T}$, note that $\boldsymbol{x}_n^{(L)}$ only depends on $x_1, \cdots, x_n$ and not on the future inputs to this time $n$. In particular, by a similar argument as in the 2-layer case (cf. Equation (85) to Equation (90)), for any $i \leqslant n - k$ we can decompose the dot-product of the key and query vectors at the $L^{\text{th}}$ layer as a bilinear form which looks like,

$$\left\langle \boldsymbol{W}_K^{(L)} \big( \boldsymbol{x}_i^{(L)} + \boldsymbol{p}_{n-i}^{(L),K} \big), \boldsymbol{W}_Q^{(L)} \boldsymbol{x}_n^{(L)} \right\rangle$$
$$= \boldsymbol{x}^T \boldsymbol{M}_{n,i}^\times \boldsymbol{y} + \boldsymbol{y}^T \big( \overline{\boldsymbol{M}}_{n,i}^\times \big) \boldsymbol{y} + (\boldsymbol{m}_{n,i}^\times)^T \boldsymbol{x} + (\overline{\boldsymbol{m}}_{n,i}^\times)^T \boldsymbol{y} + m_{n,i}^\times \triangleq f_{n,i}^{(L+1)}(\boldsymbol{x}, \boldsymbol{y}) \tag{108}$$

where $\boldsymbol{x}$ and $\boldsymbol{y}$ are defined as $\begin{bmatrix} x_n' & \cdots & x_{n-k+1}' \end{bmatrix}^T$ and $\begin{bmatrix} x_{i-1}' & \cdots & x_{i-k}' \end{bmatrix}^T$ respectively, and $\boldsymbol{M}_{n,i}^\times$ has rank at most that of $\boldsymbol{M}_n^{(L-1)}$, which is $2^{L-1}$. In particular, if $2^{L-1} \leqslant k - 2$, by Lemma 2 the proof concludes. $\qquad\square$

### E.3 The general case: Transformers with $H_\ell$ heads in layer $\ell$: Proof of Theorem 6

The $h^{\text{th}}$ head of the first layer of the attention-only transformer learns patterns of the form,

$$\widetilde{\boldsymbol{x}}_n^{(1,h)} = \sum_{i \leqslant n} \text{att}_{n,i}^{(1,h)} \boldsymbol{W}_V^{(1,h)} \text{Emb}(x_i) + \sum_{i \leqslant n} \boldsymbol{W}_V^{(1,h)} \boldsymbol{p}_{n-i}^{V,(1,h)} \tag{109}$$

$$= \sum_{i \leqslant n} \text{att}_{n,i}^{(1,h)} \left( \frac{\phi^h(0) + \phi^h(1)}{2} + x_i' \cdot \frac{\phi^h(1) - \phi^h(0)}{2} \right) + \sum_{i \leqslant n} \boldsymbol{W}_V^{(1,h)} \boldsymbol{p}_{n-i}^{V,(1,h)} \tag{110}$$

where the last equation assumes a binary input sequence, defines $x_i' = 2x_i - 1$ and uses the notation $\phi^h(0) = \boldsymbol{W}_V^{(1,h)} \text{Emb}(0)$ and $\phi^h(1) = \boldsymbol{W}_V^{(1,h)} \text{Emb}(1)$. We can further rewrite this as,

$$\widetilde{\boldsymbol{x}}_n^{(1,h)} = \boldsymbol{m}_n^{(1,h)} + \boldsymbol{M}_n^{(1,h)} \boldsymbol{x}_{1:T} \tag{111}$$

where each $\boldsymbol{M}_n^{(1,h)} \in \mathbb{R}^{d \times T}$ is rank 1 and applies a causal mask on the inputs $x_i$ for $i > n$. Recall that the output of the first attention layer applies a projection matrix on the concatenation of $\widetilde{\boldsymbol{x}}_n^{(1,h)}$ across $h \in [H_1]$ and then adds a residual connection. The output can be written down as,

$$\widetilde{\boldsymbol{x}}_n^{(2)} = \text{Emb}(x_n) + \boldsymbol{W}_O^{(1)} \begin{bmatrix} \boldsymbol{m}_n^{(1,1)} \\ \vdots \\ \boldsymbol{m}_n^{(1,H_1)} \end{bmatrix} + \boldsymbol{W}_O^{(1)} \begin{bmatrix} \boldsymbol{M}_n^{(1,1)} \\ \vdots \\ \boldsymbol{M}_n^{(1,H_1)} \end{bmatrix} \boldsymbol{x}_{1:T} \tag{112}$$

$$= \boldsymbol{m}_n^{(1)} + \boldsymbol{M}_n^{(1)} \boldsymbol{x}_{1:T}, \tag{113}$$

where,

$$\boldsymbol{M}_n^{(1)} = \left( \frac{\text{Emb}(1) + \text{Emb}(0)}{2} \right) e_n^T + \boldsymbol{W}_O^{(1)} \begin{bmatrix} \boldsymbol{M}_n^{(1,1)} \\ \vdots \\ \boldsymbol{M}_n^{(1,H_1)} \end{bmatrix}, \text{ and,} \tag{114}$$

$$\boldsymbol{m}_n^{(1)} = \left( \frac{\text{Emb}(1) - \text{Emb}(0)}{2} \right) + \boldsymbol{W}_O^{(1)} \begin{bmatrix} \boldsymbol{m}_n^{(1,1)} \\ \vdots \\ \boldsymbol{m}_n^{(1,H_1)} \end{bmatrix} \tag{115}$$

Notice that the rank of the matrix $\boldsymbol{M}_n^{(1)}$ is at most $H_1 + 1$. This is because the concatenation operation can increase the rank at most additively, and since each of the $\boldsymbol{M}_n^{(1,h)}$ matrices are rank at most 1.

Following through the proof in Appendix E.2 for the $L$-layer case, we can prove inductively that at any layer $\ell$, the output looks like,

$$\boldsymbol{x}_n^{(\ell)} = \boldsymbol{m}_n^{(\ell)} + \boldsymbol{M}_n^{(\ell)} \boldsymbol{x}_{1:T} \tag{116}$$

where the rank of $\boldsymbol{M}_n^{(\ell)}$ is $\prod_{i=1}^{\ell}(H_i + 1)$. Invoking Lemma 2, if $\prod_{i=1}^{L-1}(H_i + 1) \leqslant k - 2$, the attention-only transformer cannot realize a $k^{\text{th}}$-order induction head at layer $L$.

## F  Model architecture and hyper-parameters

The experiments were run on one $8 \times A100$ GPU node.

| Parameter | Matrix shape |
|---|---|
| transformer.wte | $2 \times d$ |
| transformer.wpe | $N \times d$ |
| transformer.h.ln_1 $(\times \ell)$ | $d \times 1$ |
| transformer.h.attn.c_attn $(\times \ell)$ | $3d \times d$ |
| transformer.h.attn.c_proj $(\times \ell)$ | $d \times d$ |
| transformer.h.ln_2 $(\times \ell)$ | $d \times 1$ |
| transformer.h.mlp.c_fc $(\times \ell)$ | $4d \times d$ |
| transformer.h.mlp.c_proj $(\times \ell)$ | $d \times 4d$ |
| transformer.ln_f | $d \times 1$ |

Table 2: Parameters in the transformer architecture with their shape.

## G  Additional experimental results

Assumption 1 suggests that the attention patterns $\text{att}_{n,i}^{(\ell)}$ in layers $\ell = 1, 2, \cdots, L - 1$, as learnt by an $L$-layer attention-only transformers may only be a function of only the position indices $n, i$. In this section we run some additional experiments to test this conjecture. We train a 2 layer attention-only transformer with $k$ heads in the first layer, on data drawn from a randomly sampled $k^{\text{th}}$-order Markov process, and focus on the learnt attention patterns as a function of in the input sequence. Figure 7 plots the results of this experiment for $k = 2$ and Figure 8 for $k = 3$. While in both cases there is some variance in the attention patterns learnt by the transformer in some of the heads, we believe that this is a consequence of the iteration budget of the transformer, and specifically the fact that even if the test loss appears to have converged, the transformer may still continue changing in the parameter space. Furthermore, when the attention patterns have some non-zero but small variance as a function of the input, a relaxation of Assumption 1, we also believe that the results we proved in Corollaries 1 and 2 and Theorem 6 should carry over approximately and leave this as an interesting question for future work. Conditional lower bounds of this nature, reliant on structural assumptions the transformer appears to demonstrate in practice are an interesting area of future research.

| Dataset | $k$-th order binary Markov source |
|---|---|
| Architecture | Based on the GPT-2 architecture as implemented in [30] |
| Batch size | Grid-searched in $\{8, 16\}$ |
| Accumulation steps | 1 |
| Optimizer | AdamW ($\beta_1 = 0.9, \beta_2 = 0.95$) |
| Learning rate | 0.001 |
| Scheduler | Cosine |
| # Iterations | Up to 25000 |
| Weight decay | $1 \times 10^{-3}$ |
| Dropout | 0 |
| Sequence length | Grid-searched in $\{32, 64, 128, 256, 512, 1024\}$ |
| Embedding dimension | Grid-searched in $\{16, 32, 64\}$ |
| Transformer layers | Between 1 and 8 |
| Attention heads | Up to $k$ |
| Repetitions | 3 |

Table 3: Settings and parameters for the transformer model used in the experiments.

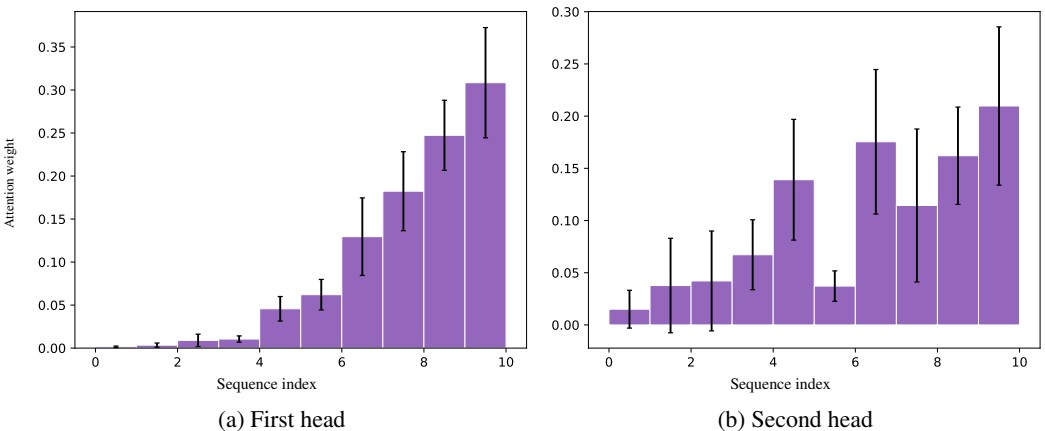

(a) First head

(b) Second head

Figure 7: Mean attention for column $n = 10$ of the two heads of the first attention layer, for a 2-layer 2-head transformer model trained on an order-3 Markov process, averaged across 100 input sequences of length 128.

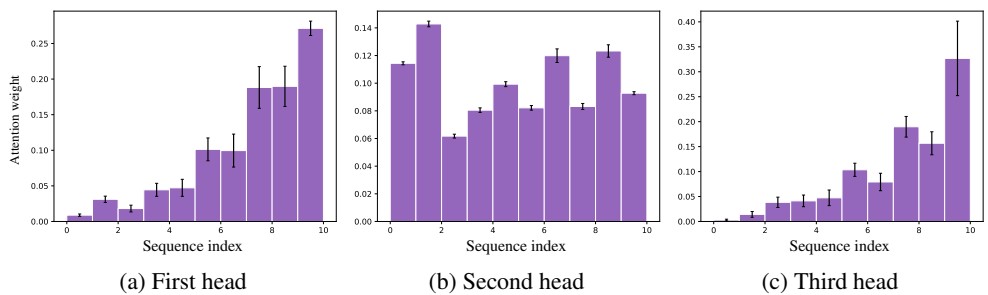

(a) First head

(b) Second head

(c) Third head

Figure 8: Mean attention for column $n = 10$ of the three heads of the first attention layer, for a 2-layer 3-head transformer model trained on an order-3 Markov process, averaged across 100 input sequences of length 128.

