# OpenReview forum: "Transformers on Markov data: Constant depth suffices"
_NeurIPS.cc/2024/Conference — NeurIPS 2024 poster_

### Official Review · Reviewer_TFSx · 2024-07-09

**Soundness:** 2
**Presentation:** 3
**Contribution:** 2
**Rating:** 5
**Confidence:** 3

**Summary:**

This paper attempts to provide a possible explanation the capability of transformer architecture for efficient next token prediction of (stationary) $k$th order Markov data. The main result is a constructive proof that a $3$-layer transformer with a single head per layer can emulate conditional $k$ grams, and it necessarily uses the non-linearities in the transformer architecture like layer-normalization. The paper also provides lower bounds on representational powers of the transformers for Markov data.

**Strengths:**

(1) The results presented in the paper serve as a crucial step towards understanding the prediction capabilities of transformers for $k$th order Markov processes with finite state spaces (of small to moderate sizes).

(2) The paper provides a novel explanation of the (possible) way in which a layer-normalization (LN) might be used in modelling $k$-grams. To my knowledge, the non-linearities are discarded in typical prior work(s).

(3) The lower bound is provided for single layer transformer (Theorem 5) is a strong evidence signifying of importance of multiple layers.

(4) The paper is well-written and easy to follow.

**Weaknesses:**

(1) In spite of solving a general and interesting problem of generative modelling using Markov structure, the evaluation is very limited. For example, I believe that the paper does not have empirical validation of $k$-gram modelling by the transformer when the size of the state space $S$ is large.

(2) The paper does not address or comment how the training data can affect the performance during testing. This step of training might be very significant when the size $S$ of the state space is high. It would be nice to see how the mechanism works when $S$ is of order of tens or a couple hundreds (my understanding is that Figure 2 is only for binary data, $S=2$).

Typo in Architecture 1: $\textbf{x}_i^{(1)} = \texttt{Emb}(x_n)$ instead of $\textbf{x}_n^{(1)} = \texttt{Emb}(x_n)$

**Questions:**

(1) I believe that the state space is binary in your evaluations $S=2$. Can you please confirm this?

(2) In Table 3 of Appendix F, the embedding dimension is mentioned to be chosen as per grid search over \{16, 32, 64\}. Should this be not the same as $d=6S+3$ as in the proof of Theorem 4 in Appendix C.3? In this case, if $S=2$, then $d=15$ is enforced. Can you provide the explanation for grid search?

**Limitations:**

(1) Although the result sheds light on the abilities of a transformer architecture in modelling conditional $k$-grams, this is only optimal for time-homogeneous stationary Markov data. Hence, the paper does not directly have a huge impact towards understanding how transformers perform on general stochastic sequences (like language tokens, or frames in a video).

(2) The constructions in the paper work only for small state spaces (tens to a few hundreds), as the embedding dimension used in the proofs scales with its size.

---

> ### Author Rebuttal · Authors · 2024-08-07
>
> We thank the reviewer for the feedback and constructive criticism. We have fixed the typo pointed out. Below we address the main weaknesses discussed in the review:
>
> ### **[W1,W2] Experiments for and learning dynamics under larger state spaces?**
>
> In the attached rebuttal pdf, we ran experiments for Markov models on state spaces of size $|S| = 5$. Here too, we show that the main phenomenon discussed in the paper ($2/3$-layer $1$-head transformers learn $k$-th order Markov processes for $k > 1$) holds true. In particular, we show that a $2$-layer $1$-head transformer learns order-$3$ Markov processes. We were unable to run this experiment for $k=4$ keeping the state space size fixed since the model longer to converge than our computational allowance would permit. When the state space becomes even larger, say on the order of tens or a couple of hundreds, it becomes much harder to train the models on reasonably large values of $k$ within the computational budget in our availability.
>
> ___
>
> ### **[Q1] Binary state space**
>
> You are correct to note that we run our experiments on binary data (to be able to scale to the largest possible values of $k$ on a computational budget).
>
> ### **[Q2] Precise dependency of embedding dimension**
>
> The experiments we run are more in line with the case of the standard transformer architecture, which corresponds to Theorem 4, where the embedding dimension scales as $O(S)$ (where the constant is implicit). In the common rebuttal, we discuss how to theoretically reduce the dependency on the embedding dimension at the cost of a small increase in the loss incurred by the model.
>
> On a separate note, there are usually a few reasons to expect that there would be some differences in theoretical and empirical findings especially when it comes to the precise dependency on quantities like the embedding dimension, etc. Empirically, transformers are trained from a random initialization and in a way where the model may be more / less efficient at realizing certain mechanisms compared to a theoretical construction. We would argue that it is in fact promising that the embedding dimension is of a similar order as what is predicted in our theorems, as it is often the case that the theoretical and empirical findings are off by several orders of magnitude.
>
> ### **[L1] Time inhomogeneous processes**
>
> This is a great question, and something we hope to study as the next step. Time homogeneity of the data model (such as with $k$-th order Markov processes) implies that it is possible to write down the estimate of the next-symbol conditional distribution using an unweighted empirical estimate. When the data process becomes time inhomogeneous, the statistical estimators would look more akin to weighted conditional empirical estimates (under assumptions on the nature of time inhomogeneity). The constructions we consider in the paper do have the ability to translate to these kinds of settings as well. However, to understand the limits of how transformers do this is a great direction for future research, and something we hope to explore going forward. It is plausible that transformers need more depth to be able to capture these kinds of processes, depending on the nature of the time inhomogeneity.
>
> ### **[L2] Constructions only apply for small state spaces**
>
> In commercial (for which metadata is available) and larger open source models, the vocabulary size has usually been in the range 30K-250K. However, the embedding dimensions have been of the scale of 1000-20,000, which is one order of magnitude smaller. It is worth pointing out that our constructions, while optimized for the dependency on the depth and the number of heads, can also be improved on the depth dependency. In the common rebuttal, we discuss how to improve the dependency of the embedding dimension from $O(S)$ to $O(\log (S)/\epsilon^2)$ where $\epsilon$ is a parameter which now captures an approximation error. Thus, even when the state space is larger, for practical ranges of $\epsilon$, the dimensionality implied by this result should be much smaller than $O(S)$.

---

> > ### Author Response · Authors · 2024-08-12
> > **Discussion period ending soon: call for response**
> >
> > Dear Reviewer TFSx,
> >
> > We sincerely appreciate the time you have taken to provide valuable feedback for our work. As we are getting closer to the end of the discussion period, could you let us know if our responses above have adequately addressed your concerns? We remain at your disposal for any further questions.
> >
> > If you agree that our responses to your reviews have addressed the concerns you listed, we kindly ask that you consider whether raising your score would more accurately reflect your updated evaluation of our paper. Thank you again for your time and thoughtful comments!
> >
> > Sincerely, The Authors

---

> > ### Comment · Reviewer_TFSx · 2024-08-12
> >
> > Thank you for the comments and clarifications. I shall retain my score as the applicability of such a theoretical statement is rather limited.

---

### Official Review · Reviewer_uqDv · 2024-07-09

**Soundness:** 3
**Presentation:** 3
**Contribution:** 3
**Rating:** 6
**Confidence:** 4

**Summary:**

This paper studies the learning process and representational power of transformers on the data sequence generated by k-th order Markov processes (or k-gram). Theoretically, this paper proved that (1) an attention-only transformer with $O(\log k)$ layers with one head for each layer can represent the conditional (in-context) $k$-gram. (2) Enhanced with MLP and layer normalization a 3-layer transformer can express the conditional $k$-gram task. They also complement their results with a lower bound for attention-only transformers/a 1-layer transformer with MLP and layer-norm with some assumptions. Detailed empirical validations are conducted to corroborate the construction (1).

**Strengths:**

1. This paper improved the construction using $k$-head, 2-layer in previous papers on the in-context $k$-gram data by constructing a $\log k$-depth, 1-head transformer. The proof technique is based on the binary-tree-like aggregation procedure of the previous $k$ tokens information, which greatly improves the memory cost of the transformer parameters. Empirically, the construction can be partially validated by experiments showing that a 3-layer transformer can learn an in-context 8-gram. Also, a lower bound result is included (with some empirically validated assumption) to make this bound tight.

2. The construction of a 3-layer transformer with the MLP and LayerNorm is quite novel. The technique is based on the unique ternary representation of integers. The ternary representation is to maintain the uniqueness of $v_i$ after normalizing the embedding vector. The role of the non-linearity in the transformer in this $k$-gram mechanism can be crucial for improving the representation power of the model, and this result can serve as an implication.

**Weaknesses:**

The two main construction results improve the previous expressivity results. However, the second theorem (Theorem 4) still lacks empirical evidence showing that the constructed solution is the minimizer that Adam/GD converges to. The previous experiments only show that $\log k$-depth transformer can somehow be learned. It is important to figure out whether the solution is only for construction or it can be actually learned in some way.

**Questions:**

1. Is it possible to add a set of experiments for the 3-layer transformer with LayerNorm and MLP trained on the k-gram model when $k>8$? Or showing that it is hard to obtain from optimization?

**Limitations:**

See weakness.

---

> ### Author Rebuttal · Authors · 2024-08-07
>
> We thank the reviewer for the positive comments, and constructive criticism. Below we address the main weakness pointed out, as well as the question asking about experiments for $k > 8$.
>
> ### **[W1] Evidence that ADAM/GD converges to theoretical construction**
>
> This is a great question. In Fig. 3 of the rebuttal pdf attached, we plot the attention patterns learnt and observe some sort of approximately exponential decay. The theoretical constructions also match this kind of exponential decay (as seen from the attention pattern in eq. (50) in the proof of Theorem 4). While this is promising, there is certainly more to explore about how and what the transformer learns. In particular, it is interesting to see how these attention patterns evolve over the course of learning before converging at these kinds of exponential patterns. This may reveal some more about how transformers learn, and get us one step closer to a theoretical understanding of the learning dynamics of transformers when exposed to Markovian data.
>
> ### **[Q1] Experiments for $k > 8$?**
>
> This was also pointed out by other reviewers, and is indeed an important point. We would be happy to include something which provides more clarification around what happens when $k=8$, however in the form we currently consider, it is not within our computational budget to scale this to $k=9,10$ or anything higher. A longer discussion as to why this kind of computational “phase transition” occurs is provided in the common rebuttal above.

---

> > ### Comment · Reviewer_uqDv · 2024-08-07
> >
> > Thank you for your detailed clarifications. I will maintain my score.

---

### Official Review · Reviewer_873s · 2024-07-12

**Soundness:** 3
**Presentation:** 3
**Contribution:** 3
**Rating:** 6
**Confidence:** 4

**Summary:**

This paper studies the ability of transformers to learn $k^{th}$ order Markov chains. They first conduct experiments showing that transformers with 2 layers and 1 head can learn Markov chains of up to order $k=4$. Similarly, with 3 layers, they can learn Markov chains of order $k=8$. Based on these observations, they show theoretical results about the representation power. Specifically, they present novel constructions to show that attention-only transformers with 1 head can learn $k^{th}$ order Markov chains with $\log_2(k+1)$ layers. On the other hand, 2-layer attention-only transformers require $k$ heads. These results show that increasing the depth is much more beneficial than increasing the number of heads. Next, the authors show that for the full transformer model, a constant depth of 3 suffices to learn $k^{th}$ order Markov chains with an embedding dimension of the order of the vocabulary size. This result reveals the benefit of non-linearities arising from layer normalization. They also present a lower bound on the depth requirement for 1-head attention-only transformers under some assumptions on the attention pattern learned by the model.

**Strengths:**

- Overall, the paper is well-written and easy to follow. The authors present nice intuitions about the constructions and the theoretical results.

- The results in the paper showcase the benefit of depth over the number of attention heads in transformers, which contributes to our understanding of transformers. The paper also provides insights into the benefit of non-linearities that arise from layer normalization in transformers, which is interesting, and helps compare attention-only transformers with standard transformers.

**Weaknesses:**

- The main weakness is that some of the statements made by the authors about their results/observations contradicting prior work *lack clarity* and are not supported by a thorough *discussion of the potential reasons for the differences*. I think these statements should be accompanied by more context. Please see the Questions section for queries/suggestions regarding this.

- There are some minor typos and grammatical errors that should be corrected. Please see the next section for details.

**Questions:**

- The statement in lines 28-31 needs further clarification:

    - It’s not clear what range of $k$ is considered in prior works. It seems that the observations are not exactly in contradiction to prior work, since a) 2-layer 1-head transformers can’t seem to be able to learn Markov chains of order $k>4$, and b) the result in Table 1 and Theorem 2 states that for order $k$, the number of heads for 2 layers is $k$, at least for attention-only transformers. Table 1 is missing a column for the number of heads needed for standard transformers with 2 layers.

    - Has prior work considered training transformers with 3 layers? How does the statement after (ii) contradict prior observations?

- Regarding the experiments, particularly Fig. 2:

    - In Fig. 2(a), the test loss gap seems a bit high. Can the authors include results for $k=1$ in this plot for a comparison?

    - Can the authors share results when training the 3-layer 1-head transformer on Markov chains of higher order $k>8$? Is there a reason why higher-order Markov chains were not considered for the experiment in Fig. 2(b)? I am curious if there is a gap between the theoretical result about the representation power (which holds for $k>8$), and the training dynamics.

- Other suggestions:

    - In the abstract, it would be good to emphasize the word ‘constant’ in line 14.

    - I suggest using $V$ instead of $S$ for the vocabulary size.

    - It would be good to include some discussion on whether similar construction techniques (for Theorem 3) have been used in the literature.

    - I suggest including some discussion on related work on the role of softmax attention, the benefit of layer normalization techniques, and the benefit of depth in neural networks.

- Typos/grammatical errors:

    - Missing citation in line 23.

    - Extra ‘studies’ in line 27. Extra ‘them’ in line 132. Extra ‘how’ in line 272.

    - Please check the phrasing in lines 220-222.

---

> ### Author Rebuttal · Authors · 2024-08-07
>
> We thank the reviewer for the detailed comments.
>
> ### **[W1] Thorough discussion for why contradictions arise in past work**
>
> As discussed in the common rebuttal, the main reason for the differences in observations compared to [1] and [2] is the fact that the models were not trained for sufficiently long to observe the increase in test loss. Please refer to Fig. 2 in the attached document where we plot this explicitly on a random $5$-state order-$3$ Markov chain. In the beginning phase of the training (iterations 1-200), the test-loss curves plateau around the bigram performance, but as we continue training, the loss continues decaying slowly. We believe that this difference only appears because the model was not trained for sufficiently long, keeping all other variables unchanged. We will include this plot in the paper.
>
> ___
>
> ### **[Q1] Range of $k$ considered in previous work**
>
> The contradiction we mention in the $2$-layer case is empirical. Prior work such as [1],[2] train $2$-layer $k$-head models on $k$-th order Markov chains for $k=1,2,3,4$. However, we are lead to believe that the $2$-layer $1$-head model was not trained for sufficiently many iterations to learn $k=2,3,4$.
>
> Fig. 2 in the attached material showcases this difference when a $2$-layer $1$-head transformer is trained on $k$-th order Markov chains on $5$ states for $k=3$. The initial phase (first ~100 iterations), the model stagnates in performance at the level of the best bigram model. When we continue training, the loss of the model starts decreasing further and breaks past this bigram phase to learn higher order conditional $k$-gram models.
>
> Finally, it’s worth noting: while we do not have a column for the $2$ layer standard transformer in Table $1$, we believe that the main $3$-layer construction can also work for $2$-layer transformers with $2$ heads in the first layer. This is because in the $3$-layer construction we consider, the first two layers of the model essentially do not interact with each other (and operate on different parts of the embedding space), they can be implemented across two different heads of the first layer. We will add this into the paper to clarify that there exists a contradiction even theoretically.
>
> **Formally:** For $2$-layer transformers with $2$ heads, prior work argues that these models can learn up to $k \le 2$. Our work shows that these models can represent much higher values of $k$ theoretically (via an extension of Theorem $4$).
>
> ### **[Q2] Clarifications regarding Fig. 2**
>
> We have included the plot for $k=1$ in Fig. 1 of the attached document in the paper (this experiment trains $k$-th order Markov chains on $k$-head $2$-layer transformers). We will include this plot in the paper as well. There is a gap in the test loss which decreases when we move to $3$-layers and $1$-head. The reasons for this small but non-negligible test loss gap are unclear - while our theoretical construction works for the $3$ layer case, it may be possible that with $2$ layers and $1$ head transformers are unable to exactly represent the conditional $k$-gram model. It may also be possible that ADAM/GD are unable to find this solution, even if it is exactly representable. Understanding the limits of how $2$-layer $1$-head models learn is an interesting question to look into.
>
> On a separate note, It is worth mentioning that the best achievable test loss will increase as we keep the sequence length fixed and increase $k$. In the learning setup we consider, the training and test data do not come from the same Markov process, since the transformer is trained on data from randomly chosen Markov processes and tested on data from another randomly chosen Markov process. So the transformer *has* to do some form of in-context learning. When the sequence length is fixed and $k$ increases, the in-context learning problem becomes harder. The model has to learn $S^k$ conditional distributions (for each possible prefix of $k$ states). So as $k$ grows, the model has less data for each of these distributions which makes the estimation task harder. This is a major reason for why the $3$-layer transformer’s loss increases a little bit when dealing with $k=8$. We were computationally bottlenecked to train on sequence lengths of up to $500$ and at this scale, the model has just about enough data to learn each of the $256$ conditional distributions. Fixing the sequence length as $500$, the models should not be expected to have small test loss as $k$ grows to $9$ and beyond.
>
> The reason we do not train on $k > 8$ is discussed in the common rebuttal. The sequence length to consider would have to be at least around $1000$ or longer for $k=9$ and $2000$ for $k=10$. The model also takes longer to train, and we were unable to train for long enough for the model to reach convergence. In the future, we hope to be able to test out the results in this paper on longer sequence lengths (thereby allowing larger values of $k$) while keeping the depth of the model and the number of heads fixed.
>
> ### **[Q3] Additional suggestions**
>
> We shall plan to incorporate the notational changes and fix the typos as suggested by the reviewer. With the additional page in the final version, we shall also include a discussion on related work on the role of softmax attention and depth which were present in existing work. Finally, we will also include a discussion on related works which consider techniques similar to the composition technique we considered in Theorem $3$.
>
> &nbsp;
>
> ___
> ### References
>
> [1] Makkuva, Ashok Vardhan, et al. "Attention with markov: A framework for principled analysis of transformers via markov chains." arXiv preprint arXiv:2402.04161.
>
> [2] Nichani, E., Damian, A. and Lee, J.D., 2024. How transformers learn causal structure with gradient descent. arXiv preprint arXiv:2402.14735.
>
> [3] Edelman, Benjamin L., et al. "The evolution of statistical induction heads: In-context learning markov chains." arXiv preprint arXiv:2402.11004.

---

> > ### Comment · Reviewer_873s · 2024-08-11
> >
> > Thank you for the detailed rebuttal and additional figures. The clarifications are helpful and I am happy to maintain my score.

---

### Official Review · Reviewer_YGsY · 2024-07-17

**Soundness:** 2
**Presentation:** 3
**Contribution:** 2
**Rating:** 6
**Confidence:** 3

**Summary:**

This paper investigates the representation capability of transformer with different number of layers or heads when learning k-th order Markov Process. They provide theoretical results demonstrating that attention-only transformers with O(log2(k)) layers can represent the in-context conditional k-th order Markov Process. This conclusion is supported by empirical results and is novel compared with previous training dynamics results. Adding LayerNorm, they prove that standard transformers with just three single-head layers can represent arbitrary order Markov processes. The paper also presents lower bounds on the size of transformers needed to represent certain attention patterns.

**Strengths:**

1. The writting is good and easy to follow. And the proof is relatively solid. Most of the intuition is clear, like using hierarchical way to construct the intermediate logits which include information of multiple tokens.
2. The main contribution of this paper is that the authors prove there exists some transformers which have fewer head and logarithm-limited layers that can learn the determined k-th Markov Process. And these results are supported by the empirical experiments, and it's more related to the real-world case. (Like we don't need too many heads)

**Weaknesses:**

1. This work analyzes the representation capability of transformer of learning Markov Process, however, there is no clear statement of error bounds in the key theorem like Theorem 4 (although there are some helpful discussions like Remark 2). And this may related to some questions like Questions-2.
2. The intuition of the proof of Theorem 5 is still not very clear, maybe adding more examples will be helpful, like how LayerNorm works.
3. Like mentioned in limitation parts, it's more focused on representation capability rather than training dynamics. The latter one may be more difficult for analyzing. More questions in Questions-1.
4. In Figure 2, the authors should better add k=8 and k=16 to further support the conclusion.
5. There is a citation typo in Line 23.
6. Maybe you can explicitly say that learn the conditional k-gram model is equivalent to letting transformer generate the same logits as that model (i.e., Eq.1), otherwise the reviewer maybe confused by the conclusion in the main theorems.

**Questions:**

1. This work is more related to representation capability of transformer, rather than training dynamics, which are the focus of related works [6], [7] mentioned in the papers (Line 114 - 133), so is it possible that the construction of k heads is more theoretically friendly than the that in your paper for analyzing dynamics like gradient flow? And what's the possible way for training dynamics under your structure?
2. It seems that you still need $\Omega(k)$ bit precision for learning $k$-th order Markov Process, but in real-world case we don't need such a high precision. Is this necessary? and is it required by the constant layers transformer setting and not required by the $O(log(k))$ attention-only setting?

**Limitations:**

The authors have mentioned their limitations in the end of paper, like they just focus on representation capability rather than training dynamics, which maybe the futural direction.

---

> ### Author Rebuttal · Authors · 2024-08-06
>
> We thank the reviewer for their detailed comments. Below we address the main questions and weaknesses:
>
> ### **[W1] Error bounds**
>
> We are happy to include a longer discussion of error bounds in the paper. To expand, suppose all the weights in the transformer model are upper bounded by $1$. When the transformer is implemented with a precision of $m$ bits, the new attention weights $\widehat{\text{att}}_{n,i}$ satisfy,
>
> $$
> \frac{1+e^{-\varepsilon}}{1+e^{\varepsilon}} \le \frac{\widehat{\text{att}}\_{n,i}}{\text{att}_{n,i}}  \le \frac{1+e^{\varepsilon}}{1+e^{-\varepsilon}}; \quad \text{where, } \varepsilon \triangleq d \cdot 2^{-m+1}
> $$
>
> This uses the fact that by truncating vectors to $m$ bits, the approximation error in the inner product of the inner product of two $d$-dimensional vectors. i.e. $\varepsilon$, is at most $d \cdot 2^{-m+1}$. In particular, simplifying assuming $\varepsilon$ is small, we get a multiplicative error of $[ 1 - \Omega (\varepsilon), 1 + \Omega (\varepsilon)]$. A similar analysis works for layer-normalization and results in a similar error scaling. Thus, within a single attention layer, we compute vectors which are within $[1-\varepsilon', 1+\varepsilon']$ where $\varepsilon' = T \cdot 2^{-m-\log(d)}$. Iterating across $L$ layers, the error incurred scales as a multiplicative $1 \pm 2^{-m-L\log(dT)}$. Thus, when the bit-complexity scales as $m \gg L \log (dT)$, the approximation error is a multiplicative polynomially small constant in $T$.
>
> There is only one caveat in case of the constant depth transformer, the weights of the transformer in each of the $3$ layers are upper bounded by $O(2^{k})$, rather than $1$. Thus, the bit precision must now scale as $k + O(\log(ST))$.
>
> We chose to de-emphasize these details in the paper to avoid unnecessarily complicated theorem statements, and to avoid detracting from the (arguably more interesting) other phenomena discussed in the paper.
>
> ### **[W2] Intuition behind Theorem 5**
>
> We have rewritten this section to make the intuitions more clear. We have added a new “proof sketch” section, clarified what the $k$-th order induction head does in this context, and emphasized how layernorm allows the transformer realize these $k$-th order induction heads.
>
> In short, layernorm allows changing attention from $\text{att}\_{n,i} \propto \exp (\langle k_i, q_n \rangle)$ to instead look like $\text{att}\_{n,i} \propto \exp ( - \| \hat{k}\_i - \hat{q}\_n \|\_2^2)$ where $\hat{k}\_i$ and $\hat{q}\_n$ are the $L_2$-normalized key and query vectors. This is nice, because if we can realize the key vectors $k_i = \sum_{j=1}^k 2^j e_{x_{i-j}}$ and the query vectors as $q_n = \sum_{j=0}^{k-1} 2^j e_{x_{n-j}}$, the attention is maximized if and only if $\hat{k}_i = \hat{q}_n$, which we can prove occurs only when $x\_{i-1} = x\_n, \cdots, x\_{i-k} = x\_{n-k+1}$ (using the fact that the binary representation of a number is unique). This realizes a $k$-th order induction head.
>
> ### **[W3] Representation vs Training dynamics**
>
> In this paper, we focus on the representation capacity, which itself turns out to be a complicated phenomenon. On the optimization side, there are indeed many questions open. Unfortunately, even in the simplest settings (1 layer transformers), the analysis of gradient descent on transformers trained on Markovian data is incredibly complex. The rigorous analysis in [Makkuva et al] is over 70 pages long for this case. Extending to higher depth, is incredibly non-trivial. [Lee et al] discusses this case, however, their analysis is only amenable under very strong assumptions i.e., the “reduced model” of transformers. Even with these assumptions, the analysis is over 60 pages long. We hope to convince the reviewer that while training dynamics is indeed an interesting and timely question, it certainly requires a dedicated effort to be able to analyze, and would constitute a separate work of its own.
>
> ### **[W4] Experiments on higher values of $k$**
>
> Discussed in the common rebuttal.
>
> ___
>
> ### **[Q1] Learning dynamics**
>
> At a high level, yes, we believe that the k-head mechanism discussed in prior works may be more amenable for analysis. When the depth is constant, the transformer is forced to be more creative in the ways it learns the $k$-th order chain, and this leads to more intricate mechanisms. However, unless simplifying assumptions are imposed, we believe that understanding learning dynamics falls into the realm where current tools in optimization theory are not strong enough to be able to answer.
>
> ### **[Q2] $\Omega(k)$ bit precision for learning $k$-th order Markov processes**
>
> One may consider an extension of $k$-th order Markov processes to the case where the conditional distribution depends on a sparse subset (of size $p$) of the previous symbols observed. The conditional distribution has a low degree of dependency on the past, even though it is not the immediately preceding $p$ symbols. While these kinds of processes are special case of $k$-th order Markov processes, modeling them as such may necessitate a very large value of $k \gg p$, since the there may be dependencies on symbols which appeared long ago.
>
> Our constant depth constructions work even in this setting, with the bit-complexity scaling with $p$ and not with $k \gg p$. This analysis shows that transformers don’t require high bit-precision as long as the degree of dependency of the conditional distribution on the past is small. In practice, this is often the case.
>
> On a separate note, our constant depth constructions require $\Omega(k)$ bit complexity, but the embeddings are now $O(S)$ dimensional, rather than being $\Theta(Sk)$ dimensional in the attention-only setting. Thus, the total number of bits in each embedding vector is the same in both cases. We believe that it is possible to reduce the bit complexity of the constant depth construction at the cost of making the embeddings $\Theta (Sk)$ dimensional.

---

> > ### Author Response · Authors · 2024-08-12
> > **Discussion period ending soon: call for response**
> >
> > Dear Reviewer YGsY,
> >
> > We sincerely appreciate the time you have taken to provide valuable feedback for our work. As we are getting closer to the end of the discussion period, could you let us know if our responses above have adequately addressed your concerns? We remain at your disposal for any further questions.
> >
> > If you agree that our responses to your reviews have addressed the concerns you listed, we kindly ask that you consider whether raising your score would more accurately reflect your updated evaluation of our paper. Thank you again for your time and thoughtful comments!
> >
> > Sincerely, The Authors

---

> > > ### Comment · Reviewer_YGsY · 2024-08-12
> > >
> > > Thanks for your reply, and I think most of my concerns have been solved and I will raise my score to 6

---

> > > > ### Author Response · Authors · 2024-08-12
> > > > **Thank you**
> > > >
> > > > We are grateful for the reviewer providing valuable feedback and their reconsideration of the score!

---

### Official Review · Reviewer_DmcT · 2024-07-19

**Soundness:** 3
**Presentation:** 3
**Contribution:** 3
**Rating:** 6
**Confidence:** 3

**Summary:**

The paper studies the representation capacity of transformers in in-context learning of order-$k$ Markov chains. First, the authors theoretically show that $O(\log (k))$ layers are sufficient to represent $k$-th order induction heads in attention-only transformers. The paper also demonstrates the benefit of non-linearities, such as layer-norm, by showing that a slightly modified transformer (modified from the original architecture) with constant depth is also sufficient to represent the in-context conditional distribution.

**Strengths:**

Given the prevalence of in-context learning, I believe understanding the ICL of Markov chains is a very important problem, not only because of the Markovian nature of language but also the simplicity and control it provides in a synthetic setup. This is a good work that builds on recent research on ICL of Markov chains by showing that in a slightly modified transformer, scaling of heads with the order of the chain ($k$) is not necessary to represent $k$-th order induction heads. The lower bound (albeit contingent on the $k$-th order induction head assumption) also provides useful insights. I really enjoyed reading the paper; it was very clearly written and easy to follow. Generally speaking, all the ideas are laid out very clearly, with intuitive explanations that follow before and after the theorems.

**Weaknesses:**

1. It would be beneficial to compare single head results with multiple heads (e.g., in Figure 2a, $2$ heads for $k=2$ and $4$ heads for $k=4$) as a sanity check, given previous works [1] that highlight the multi-head requirement for learning order-$k$ induction heads. I mention this because, to me, it seems like in Figure 2a, the loss isn’t approaching zero unlike in Figure 2b.


2. The work mentions "long training" to observe contradictory results (2-layer single head tfs being able to learn up to order-4 MCs in-context). How long are you training compared to [1]? How significant is this effect, and what impact do you think it has in terms of the underlying mechanism being picked up? I think this needs to be discussed further.


[1] Benjamin L. Edelman, Ezra Edelman, Surbhi Goel, Eran Malach, and Nikolaos Tsilivis. The
evolution of statistical induction heads: In-context learning markov chains, 2024.

**Questions:**

1. What do you think is the reason 2-layer, 1-head transformers can learn up to order-$4$ MC (if they do, see Weakness 1)? What mechanism do you think is inherently being picked up? Is it learning order-$k$ induction heads or some alternate mechanism? I ask because if there is an alternate mechanism, it could be useful for the lower bound (Thm. 6) in establishing if "the lower bound representing $k$-th order induction heads implies an unconditional lower bound."

2. Regarding Theorem 4, even though the embedding dimension requirement is reduced compared to attention-only transformers (Thm. 2, Thm. 3), it still feels quite large compared to what we see in practice. Any comments on this? How tight do you think this is?

**Limitations:**

Yes, the authors have discussed the limitations separately in the Appendix.

---

> ### Author Rebuttal · Authors · 2024-08-06
>
> We thank the reviewer for the comments and questions about the paper. Below we address the main questions and weaknesses raised:
>
> ### **[W1] Comparing single vs. multi-head**
>
> In the attached material, please refer to Fig. 1, which we will add into the paper in the subsequent version (as the new Fig. 1). This plot discusses how the multi-head transformer learns, while the number of iterations to converge to optimality is of the same order, the test-loss these models achieves is nearly $0$. In comparison, the single head $2$-layer model has non-zero but small test-loss, while the single head $3$-layer model has even smaller test loss. At this scale, we did not distinguish between the small loss of the transformer vs. negligible loss incurred by the multi-head setup. It is indeed plausible that with better choices of hyperparameters (optimizing over embedding dimension, for instance), we may see the delta become even smaller.
>
> In our evaluation setup, the transformer does not know what the distribution of samples at test-time are (only that they are Markovian). Thus, any model which achieves low test-loss *has to* resort to something like an in-context estimate. We were computationally bottlenecked to train transformers on sequence lengths of up to around $500 \approx 2^9$ for the number of iterations required for the test-loss to begin to converge. At this scale, the model has “just” enough data to be approximately able to estimate order-8 Markov chains in context (i.e., having to approximately estimate $2^8 \approx 250$ conditional distributions on $\{ 0,1\}$). At this scale, if were to increase $k$ the model would stop performing well, and more importantly, no in-context estimator could work at this scale. Information theoretically, there is just not enough data in a test-sequence to be able to estimate the next sample distribution, even approximately.
>
> **TLDR;** When the sequence length is around $500$, $k=8$ is the point where we would start to see degrading performance (for any model, and not just a transformer) by virtue of approaching the regime where we don’t have enough samples to estimate conditional probabilities from the test-sequence accurately.
>
> ### **[W2] Long training to observe contradictory results**
>
> This is discussed in the common rebuttal above.
>
> ___
>
> ### **[Q1] Why do 2-layer 1-head transformers learn order-4 Markov chains?**
>
> This is a great question. While our construction shows that the standard $3$-layer transformers with $1$ head are able to represent $k$-th order induction heads, it may be possible that at an even smaller scale (say $2$-layer with $1$ head) that transformers are still able to represent the in-context conditional $k$-gram with some amount of error. It is also possible that at an even smaller scale, (say $2$-layer $1$-head transformers for sufficiently large $k$, or if the embedding dimension is reduced even further) that induction heads are no longer representable at all. Understanding this regime presents new avenues of research: here the transformer may still achieve low loss, while “provably” not being able to use the induction head mechanism to achieve this loss.
> It’s worth pointing out the case of $1$-layer $1$-head transformers which were studied in [1]. The authors show that transformers are still able to learn Markov chains, but under this extreme size constraint, learning does not happen in a transductive way. At this scale, the transformer does not do in-context learning, rather learns the parameters of the Markov chain from which data is generated directly. This is a slight departure from our setting, since learning the parameters is infeasible when the training and test data come from different distributions (such as two different randomly sampled Markov chains).
>
> ### **[Q2] Precise dependency on embedding dimension**
>
> This is discussed in the common rebuttal above.
>
> &nbsp;
> ___
> ### References
> [1] Makkuva, Ashok Vardhan, et al. "Attention with markov: A framework for principled analysis of transformers via markov chains." arXiv preprint arXiv:2402.04161 (2024).

---

> > ### Comment · Reviewer_DmcT · 2024-08-13
> >
> > Thank you for the detailed responses and additional figures. I have read other reviews and responses as well, and I am happy to retain my rating.

---

### Author Rebuttal · Authors · 2024-08-07

## **Common Rebuttal**

We thank all the reviewers for taking the time to go through our paper and suggest constructive criticism. Please find attached a pdf containing additional plots.

Below we address some common points raised by multiple reviewers.

### **Long training reveals contradictions with prior work**

We point the reviewers to Fig. 2 in the rebuttal pdf. Here, we clearly observe the stages of learning for a $2$-layer transformer. Initially (in the first ~200 iterations) the model stagnates and this is around the loss of the best bigram model (which is what previous work claimed these models could learn). As we continue training, we see that the loss starts decaying but over a much slower time-scale. The model indeed requires around 20K iterations to get close to the optimal loss. The closest related phenomenon to why the model performance appears to improves very slowly is that of grokking: zooming into any interval, it may appear that the model has converged, however this only appears to be the case as the parameters of the model may be in a region where the loss landscape is very flat.

In the rebuttal pdf, we also point the reviewers to Fig. 1. The authors of [1] also present similar results, but it appears that their experiments requires more samples / optimization cycles than in our paper. Here, we observe that $4$-head transformers with $2$ layers need around 3K iterations to learn $k$-th order Markov chains for $k=4$. In contrast, according to Fig. 2 in the main paper, $2$-layer transformers with $1$ head trained on the same data processes require around 7K iterations.

### **Running our experiments on higher values of $k$**

Running our experiments in Fig. 2 on higher values of $k$ is certainly important. However, there is a computational tradeoff which we were not able to navigate when $k$ grows to be larger than $8$. We were computationally bottlenecked to train transformers on sequence lengths of up to around $500 \approx 2^9$ for the number of iterations required for the test-loss to begin to converge. At this scale, the model has “just” enough data to be approximately able to estimate order-8 Markov chains in-context. Note that the model has to approximately estimate $2^8 \approx 250$ conditional distributions to learn order-8 Markov chains. At this scale, if were to increase $k$, keeping the sequence length fixed, the model would stop performing well. This is the scale at which no in-context estimator could work. Information theoretically, there is just not enough data in a test-sequence to be able to estimate the next sample distribution, even approximately. In order to test our results, say for $k=10$, we would have to evaluate on transformers of sequence length $\approx 2000$ (4x blowup). While the forward/backward passes naively takes 16x longer, the transformer also seems to require many more iterations to optimize the loss to low error. These reasons make it prohibitive to evaluate on larger values of $k$.

### **Precise dependency on embedding dimension, especially when $S$ is large**

In practice, the embedding dimension of transformer models like GPT-2/3 and LLama-2/3, and plenty of other open source models (for which data is available), the embedding dimension falls roughly in the range (1000-20,000). This is an order of magnitude smaller than the vocabulary size of these models, which is often in the range (30,000-200,000). It should be possible to improve our results in the dependency on the embedding dimension, which we didn’t fully optimize for (in comparison with the number of heads and depth). At a high level, the reason the embedding dimension in our Theorems  and ___ scales as $O(S)$ is because the transformer stores information about the statistics of each symbol along orthogonal components of the embedding vector. The orthogonality of these components prevents information about the statistics of any one symbol from affecting those of another symbol.

It remains to be formally verified, but we believe that all of our constructions, as well as the mechanisms empirically learnt by transformers are robust to errors which may appear even when information about the statistics of each symbol does not appear in exactly orthogonal components of the embedding vector, but in *approximately* orthogonal components. By allowing for approximate orthogonality, the transformer can be much more efficient about its use of the embedding space: the Johnson Lindenstrauss theorem states that there are $S$ approximately orthogonal (pairwise inner products $\in [-\epsilon,\epsilon]$) vectors in $\mathbb{R}^p$ even when $p = O(\log (S)/\epsilon^2)$ is much smaller than $S$. In principle, it suffices to choose the embedding dimensions to scale roughly as $\log (S)/\epsilon^2$ to have an approximate version of Theorem $4$, paying a error scaling with $\epsilon$.

Finally, it’s worth mentioning that for practical ranges of $\epsilon$ and $S$, this number ($\log(S)/\epsilon^2$) can indeed be much smaller than what is predicted by what is in the current version of Theorem $4$.

We are happy to include this discussion in the paper to provide more context about the embedding dimension.

&nbsp;

___

### References
[1] Edelman, Benjamin L., et al. "The evolution of statistical induction heads: In-context learning markov chains." arXiv preprint arXiv:2402.11004.

---

### Comment · Area_Chair_jrDo · 2024-08-11
**Dear reviewers, please read and respond to authors' rebuttal (if you haven't done so)**

Thanks!

Your AC.

---

### Decision · Program_Chairs · 2024-09-25

**Decision:**

Accept (poster)

**Comment:**

The paper shows expressibility and empirical results of Transformers when the input context follows k-th order Markov chains (i.e., the in-context learning setting), and characterizes the number of layers required to learn high-order chains from the theoretical point of view, showing surprising capability of Transformer of log_2(k) layer to encode k-th order Markov chains. Reviewers all agree the direction addressed by the work is important and novel.